# WorldGUI: An Interactive Benchmark for Desktop GUI Automation from Any Starting Point

## Abstract

GUI agents have achieved outstanding performance in GUI element grounding. However, planning remains highly challenging, especially due to the sensitivity to the initial state of the environment. Specifically, slight differences in the initial state-such as the target software not being open or the interface not being in its default state, often lead to planning errors. This issue is widespread in real application scenarios, but existing benchmarks fail to evaluate it. To address this gap, we introduce WorldGUI, a comprehensive GUI benchmark containing tasks across ten widely used desktop and web applications (e.g., PowerPoint, VSCode, Acrobat), each instantiated with diverse initial states to simulate authentic human–computer interactions. Complementing this, we propose WorldGUI-Agent, a universal framework that unifies three core modules: Planner-Critic for high-level plan refinement, Step-Check for intermediate verification, and Actor-Critic for action-level optimization to proactively detect and correct errors. Experimental evaluation shows that WorldGUI-Agent outperforms the outstanding existing model (Claude-3.5-Sonnet CCU) by 12.4% in success rate on WorldGUI, and achieves a 31.2% overall success rate on WindowsAgentArena, surpassing the prior state-of-the-art by 11.7%. Our analysis further reveals that dynamic augmentation tasks and desktop environments pose substantial hurdles, underscoring the necessity of adaptive planning and feedback-driven execution for advancing real-world GUI automation.

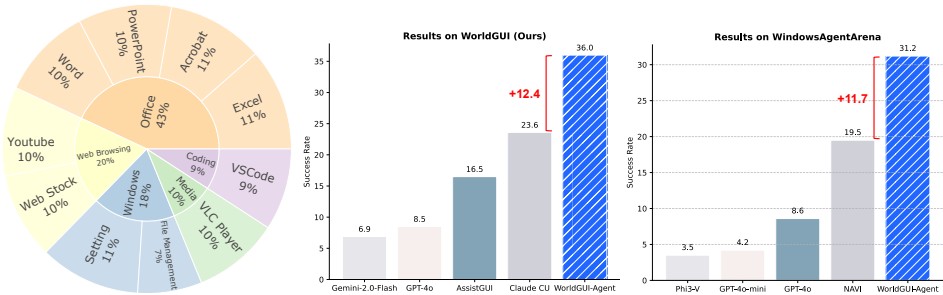

Figure 1: Software taxonomy of WorldGUI and the performance comparison of GUI agents. The left shows 5 main groups and 10 software in our WorldGUI. The right shows that WorldGUI-Agent surpasses previous SOTA GUI agents significantly.

# 1 Introduction

Graphical User Interface (GUI) automation has become a prominent research area, driven by the need to enhance user productivity. This domain encompasses software usage, file management, office design, coding, and web browsing. Building upon Multimodal Large Language Models (MLLMs) such as GPT-4o (OpenAI, 2023) and Claude-3.5-Sonnet (Anthropic, 2024), GUI agents have the potential to solve various computer tasks to avoid repetitive work or as an AI assistant to enhance productivity efficiency.

Figure 2: **WorldGUI.** Left: WorldGUI creates pre-actions for each meta task, leading to different initial states. It successfully reflects the real-world human-computer interaction process. Right: components in WorldGUI.

GUI automation operates in a *dynamic* environment, which goes beyond the traditional computer vision tasks like image recognition (He et al., 2016) and visual question answering (Antol et al., 2015; Goyal et al., 2017). However, current online GUI benchmarks such as WebArena (Zhou et al.), WebVoyager (He et al., 2024), and WindowsAgentArena (Bonatti et al., 2024) do not capture this dynamism. Currently, most GUI benchmarks (Xie et al., 2024; Bonatti et al., 2024; Gao et al., 2024; Zhou et al.; Koh et al., 2024; He et al., 2024) focus on initial and final states, measuring success rates but overlooking the state variety in real GUI scenarios. These benchmarks often ignore situations where: (1) The software interface is not in its default state. (2) The human-computer interactions may start from the intermediate state of a specific task. (3) Differences in agent robustness, where agents with the same low success rate (e.g., 20%) may vary in their ability to self-reflection, but these abilities cannot be measured in a *static* setting. As a result, these benchmarks fail to comprehensively assess the GUI agents.

In this paper, we take the first step toward comprehensive GUI evaluation by designing GUI tasks with various initial states. We consider that the testing process of WorldGUI can be featured: **(1) Intermediate Starting States**: Real user interactions with GUI assistants do not always begin from default initial conditions, allowing tasks to start from intermediate states where users may seek assistance at any point. **(2) Contextual Variability**: In some cases, tasks may originate from entirely different contexts or interfaces, requiring the agent to adapt by modifying existing plans or introducing new steps to ensure task execution. By incorporating these situations into the benchmark design, WorldGUI better mirrors real-world GUI interactions, enabling a more accurate and thorough assessment of GUI agent capabilities. Specifically, WorldGUI embraces 10 widely-used desktop applications with 611 tasks in total. For each task, we create a user query, an instructional video, and the corresponding project file. We engaged four trained annotators skilled in using these applications for annotation. To simulate the dynamic testing scenarios, we demonstrate each task to obtain ground-truth (GT) plans and then conduct the augmentations for each task using pre-actions.

In addition, we introduce a new GUI agent framework, WorldGUI-Agent, which builds upon critical thinking design principle, an aspect less emphasized in previous GUI agents (Hong et al., 2024; Cheng et al., 2024; Lai et al., 2024; Agashe et al., 2025a; Wu et al., 2024). In dynamic GUI environments, application settings may not be in default configurations. This unpredictability requires agents to have the essential ability to detect and adapt to such changes to ensure task accuracy. Through our analysis of real-world GUI scenarios, we identify three design principles for GUI agents: **(1) Post-Planning Critique**, **(2) Pre-Action Validation**, and **(3) Post-Action Evaluation**. We argue that these components are fundamental and universal for GUI agents.

To summarize, our key contributions are the following: **(1)** We are the first to stress the dynamic testing processes in the online GUI testing and propose a new benchmark called WorldGUI; **(2)** We introduce WorldGUI-Agent, a fundamental and universal GUI framework that incorporates critical thinking into the overall agent design, providing valuable insight and guidance for future development; **(3)** We explore the essential property of critical thinking in GUI agents and empirically show that critical thinking is extremely useful for handling GUI tasks (see Figure 1).

## 2 WORLDGUI BENCHMARK

### 2.1 TASK FORMULATION

**GUI Automation Definition.** The GUI automation task can be considered a partially observable Markov decision process (POMDP) $(\mathcal{S}, \mathcal{O}, \mathcal{A}, \mathcal{T}, \mathcal{R})$ with state space $\mathcal{S}$, observation $\mathcal{O}$, action space

Table 1: **Comparison with other interactive GUI benchmarks. WorldGUI** is a unique benchmark that embraces diverse initial states and better reflects the authentic interactions in GUI scenarios. Env?: Indicates whether an environment is required to be deployed.

| Benchmarks | Softwares | Tasks | Platform | Env? | Inst. | Video? | GT Plan | Diverse Init. State? | Contextual Variability? |
|---|---|---|---|---|---|---|---|---|---|
| WebArena (Zhou et al.) | 6 | 812 | Web | Yes | ✗ | ✗ | ✗ | ✗ | ✗ |
| VisualWebArena (Koh et al., 2024) | 3 | 910 | Web | Yes | ✗ | ✗ | ✗ | ✗ | ✗ |
| WebVoyager (He et al., 2024) | 15 | 643 | Web | Yes | ✗ | ✗ | ✗ | ✗ | ✗ |
| AutoDroid (Wen et al., 2024) | 13 | 158 | Android OS | Yes | ✗ | ✗ | ✗ | ✗ | ✗ |
| AndroidWorld (Rawles et al., 2024) | 20 | 116 | Android OS | Yes | ✗ | ✗ | ✗ | ✗ | ✗ |
| AgentStudio (Zheng et al., 2025) | 9 | 205 | Desktop + Web | Yes | ✗ | ✗ | ✗ | ✗ | ✗ |
| Mobile-Eval (Wang et al., 2024) | 10 | 30 | Android OS | Yes | ✗ | ✗ | ✗ | ✗ | ✗ |
| APPAgent (Zhang et al., 2023) | 10 | 50 | Android OS | Yes | ✗ | ✗ | ✗ | ✗ | ✗ |
| OSWorld (Xie et al., 2024) | 10 | 369 | Desktop | Yes | ✗ | ✗ | ✗ | ✗ | ✗ |
| AssistGUI (Gao et al., 2024) | 9 | 100 | Windows | No | ✓ | ✗ | ✗ | ✗ | ✗ |
| WindowAgentArena (Bonatti et al., 2024) | 11 | 154 | Windows | Yes | ✗ | ✗ | ✗ | ✗ | ✗ |
| **WorldGUI** | 10 | 611 | Win. + Web | No | ✓ | ✓ | ✓ | ✓ | ✓ |

$\mathcal{A}$, transition function $\mathcal{T}: \mathcal{S} \times \mathcal{A} \to \mathcal{S}$, and reward function $\mathcal{R}: \mathcal{S} \times \mathcal{A} \to \mathbb{R}$. In our setting, given a natural language query $q$, eg., *Format the slide background with gradient fill* that describes a specific task in high-level, along with an instructional video $v$ as a supplement that more detailed illustrates how to complete it, the agent first get the observation $o_t \in \mathcal{O}$ from the state $s_t \in \mathcal{S}$ in the execution environment and then generate the executable action $a_t \in \mathcal{A}$, resulting in a new state $s_{t+1} \in \mathcal{S}$ and a new observation $o_{t+1} \in \mathcal{O}$. The process repeats until the task is finished or fails. The reward function $\mathcal{R}: \mathcal{S} \times \mathcal{A} \to [0, 1]$ here returns a binary integer at the final step ,indicating the task completion status.

**WorldGUI Task Definition.** As illustrated in Figure 2, to achieve state diversity within each task, we generate various initial states that converge to the same final state, resulting in distinct ground truth (GT) plans for each case. This is accomplished through the use of **pre-actions**, which consist of a sequence of executable code to initialize tasks from different initial states. With the augmentation of initial states, WorldGUI is capable of mimicking the different testing scenarios. We additionally summarize the differences between WorldGUI and other close interactive benchmarks in Table 1.

**Observation Space.** The observation space $\mathcal{O}$ indicates the information of the operating system (OS) available to the agent in each state $s_t$. In this paper, we follow the previous work of Assist-GUI (Gao et al., 2024), encompassing two types of information: metadata $m_t$ from the application and screenshot $V_t$ of the current state $s_t$. The metadata mainly includes the layout of panels and UI trees. The screenshot $V_t$ offers holistic visual information of the current state used for planning and action generation.

**Action Space.** Our action space includes all raw mouse and keyboard actions, such as left-click, right-click, double-click, drag, keystrokes, and key combinations for shortcuts, among others. Mouse-related actions also specify the target position in the pixel space of the observed screenshot. To ensure a universal and comprehensive representation of actions, we adopted the widely used Python library, PyAutoGUI[1], for controlling mouse and keyboard inputs. Each action is represented using the syntax `action_type(arguments)` as in Table 2.

Table 2: The action types and their example in WorldGUI.

| Action Type | Example |
|---|---|
| Mouse Movement | `moveTo(120, 200)` |
| Mouse Clicks | `click(200, 300)` |
| Keyboard Type | `write('classes')` |
| Hotkey | `hotkey('ctrl', 'a')` |
| Scrolling | `scroll(-100)` |
| Drag | `dragTo(120, 220, 2)` |
| Mouse Down and Up | `mouseDown(); mouseUp()` |
| Press Keys | `press('delete')` |
| Key Down and Up | `keyDown('shift')` |

## 2.2 DATA SOURCE

WorldGUI consists of a broad spectrum of widely-used desktop applications, which can be categorized into five main groups: *(i)* Office, includes PowerPoint, Word, Excel, and Adobe Acrobat; *(ii)* Windows Usage, includes System Settings and File Management; *(iii)* Web Usage, includes the configuration of Youtube and website operations; *(iv)* Coding, focus on the customization, configuration and editing of Visual Studio Code (VSCode); *(v)* Media, operating VLC player for video editing and creation.

---

[1]https://pyautogui.readthedocs.io

## 2.3 PIPELINE OF DATA CONSTRUCTION

We engaged four annotators and developed the necessary scripts to structure and format the data. Additionally, to facilitate ground truth (GT) plan generation and pre-action generation, we implemented simple agent-based methods to collect the relevant data. The overall data construction pipeline comprises six steps, as detailed below.

**Raw Video Collection.** We collect raw videos from the YouTube website as there are a lot of high-quality tutorials for desktop applications with high views. For each software, we ask the annotators to watch the videos first and download them via the diversity of software usage.

**Instruction Video Preparation.** After obtaining the raw videos, we write the script codes to cut the lengthy and noisy videos into the sub-clips (30 seconds to 3 minutes) that serve as the instructional video.

**User Query Generation.** After obtaining the instructional videos, annotators are asked to manually write user queries corresponding to each video. For example, a user query for a task involving File Explorer might be: *"Please compress the project.mp4 into an MPEG-4 file optimized in 1080p."*

**Project File Preparation.** Following the AssistGUI (Gao et al., 2024), we create the project file for each task to ensure reproducibility without relying on resource-intensive virtual machines (Xie et al., 2024) or Docker environments (Bonatti et al., 2024). This approach guarantees that the testing process begins from a consistent state. When combined with pre-actions, it enables augmentation of the same task with various initial states.

**GT Plan Generation.** We write the script to accept user query $q$ and instructional video $v$ as input and generate the raw plans by agent (powered by GPT-4o). Since the raw plans are not flawless, annotators are asked to watch the videos and manually execute the tasks following the raw plans. During this process, annotators edit the plans to correct any inaccurate steps or descriptions, ultimately producing the finalized GT plans.

**Pre-Actions Generation.** To vary the task, we propose introducing pre-actions before the task begins. These pre-actions are created by annotators and involve corresponding scripts and agents. They are written in Python code, for example: `from pyautogui import click, rightClick\n rightClick(800,400)`. The pre-actions primarily serve two purposes: **1) Simulating Intermediate Task States**: Pre-actions can complete specific steps of a task, creating a starting point from an intermediate state. This approach addresses scenarios where users may invoke GUI assistant at any time. For example, if the task involves opening a dropdown menu, the pre-action may pre-open the menu. If the agent fails to recognize this precondition and follows its plan to click the menu again, it might inadvertently close the menu, causing task failure. **2) Introducing Diverse Initial Context States**: Pre-actions can also introduce variations in the initial state, such as opening random tabs or settings. This ensures that the starting state is unconventional, challenging the agent to adapt by modifying its plan or adding necessary new steps. See example in Figure 8.

## 2.4 EVALUATION

WorldGUI employs an execution-oriented evaluation approach followed by AssistGUI (Gao et al., 2024) and WindowsAgentArena (Bonatti et al., 2024) by utilizing post-processing scripts to assess task completion. Specifically, for tasks like Office work and Web Browsing, we adopt exact matching to compare the differences between the ground-truth (GT) screenshots and the final screenshots. For tasks like File Management, which would produce new folders or change the locations of files, etc. We create the shell script to check the status of files.

## 2.5 DATA STATISTICS

WorldGUI compiles GUI tasks from 10 widely used applications on the Windows platform, including productivity software such as PowerPoint, Word, Excel, and VSCode. A total of 111 meta tasks were collected from these applications, with each task being augmented 5 times based on the task's functionality, resulting in 500 augmented tasks. In total, WorldGUI comprises 611 tasks, and every task has almost 6 variation instances, which is capable of reflecting the real-world interactions of the GUI environment. See the details in Table 11 (more details in the Supplementary Material).

**1. Planner**
Generate the plans based on current state

Task 1: Adjust text wrapping around a Word table
Subtask 1: Click within the table to select it.
Subtask 2: Click on the 'Layout' tab in the toolbar.
Subtask 3: Click 'Properties' from the options available.
...

**Desktop Environment**

**User Query + Instructional Video**

**Observe**

**2. Planner-Critic**
Assess the correctness of plans generated by Planner

**Observe**

**5. Actor-Critic**
Assess the success of the last action
If wrong, correct the action

Task 1: Adjust text wrapping around a Word table
**Subtask 1: Click on the 'Table Layout' tab in the toolbar.**
Subtask 2: Click within the table to select it.
Subtask 3: Click 'Properties' from the options available.
...

**Observe**

**Observe**

**Act**

**Feedback**

**3. Step-Check**
Check the task completion and redundancy

**4. Actor**
Generate action represented by code
Correct the action with critic feedback

Figure 3: **WorldGUI-Agent**. The Planner module receives the user query and an instructional video as input and generates an initial plan. This plan is then refined and executed step by step. Before each step is passed to the Actor module, it undergoes a Step-Check. After the Actor produces an action, the Actor-Critic module iteratively verifies the completion of the action and makes corrections.

# 3 WORLDGUI-AGENT: THINKING BEFORE DOING

In this section, we introduce an universal GUI framework **WorldGUI-Agent** with a core and essential designing principle: *critical thinking*, which is vital for designing GUI agents capable of handling dynamic environments that have been overlooked in prior GUI agents (Hong et al., 2024; Cheng et al., 2024; Lin et al., 2024; Zhang et al., 2023; Agashe et al., 2025a). The WorldGUI-Agent includes the **five fundamental but essential components** as in Figure 3 and an **Interaction reasoning loop** detailed in Algorithm 1. We summarize our critical designs in the following:

• **Post-Planning Critique:** After the planning phase, a critique module verifies and, if necessary, self-corrects the generated plans to ensure their accuracy.

• **Pre-Action Validation:** Before executing each subtask, a validation module determines whether the subtask should be executed. This step is crucial, as the current GUI environment may indicate that the subtask is unnecessary or requires modification to align with the current state.

• **Post-Action Evaluation:** After each action execution, a mechanism evaluates whether the action was successfully completed before proceeding to the next subtask.

These critique designs ensure the reliability and adaptability of WorldGUI-Agent in complex GUI environments.

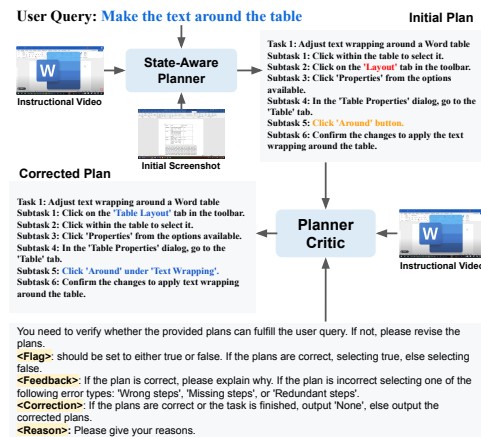

Figure 4: **State-Aware Planner and Planner-Critic.** The Planner generates an initial plan. Then, the Planner-Critic provides necessary corrections.

## 3.1 STATE-AWARE PLANNER

The State-Aware Planner processes the instructional video $v$ and the user query $q$ generates an initial plan as shown in the left of Figure 4. We use the speech recognition model Whisper (Radford et al., 2023) to translate the video $v$ into the subtitle and then send it to the MLLM for task planning. The task plan is hierarchically structured as $p = [p_1, p_2, ..., p_N]$ where $p_i$ is a text string describing the $i$-th milestone of the task. Under each $p_i$, there is a list of subtasks $[S_1^i, S_2^i, S_N^i]$, where $S_j^i$ is the $j$-th subtask in the $i$-th milestone. To ensure the produced plans fit the GUI environment, we propose incorporating an initial screenshot $V_0$ to represent the current state. This additional context allows the agent to output plans that align with the actual state. For example, if the instructional video suggests clicking on the "Layout" tab in the Word application, but the current state (as indicated by the screenshot) shows that the "Layout" tab is already selected, there is no need to perform this action again. By utilizing the visual information

from the screenshot, the State-Aware Planner can modify the plans accordingly, rather than strictly following the guidance in the instructional video or the existing knowledge from backbone MLLMs. It also avoids the occlusion issue when not seeing the screenshot.

## 3.2 PLANNER-CRITIC

**Post-Planning Critique.** The goal of the Planner-Critic is to assess the correctness of the initial plans generated by the State-Aware Planner and provide corrections if needed. This module is designed to ensure the accuracy of the plans while leveraging the self-reflection capabilities of MLLMs. As illustrated in Figure 4, for each Initial Plan, the output consists of four components:

(1) <Flag>: Indicates whether the Initial Plan is correct.

(2) <Feedback>: Identifies the error type, categorized into one of three options: "Wrong Steps," "Missing Steps," or "Redundant Steps."

(3) <Correction>: Provide the corrected plans if the Flag indicates that the Initial Plan is incorrect.

(4) <Reason>: In addition to giving the corrected plans, we force the model to give the reasons. As previous studies (e.g., CoT (Wei et al., 2022), Deepseek-R1 (DeepSeek-AI et al., 2025)) demonstrate that generating reasoning steps along with the answer would enhance the performance.

## 3.3 STEP-CHECK

**Pre-Action Validation.** After the plan assessment, a navigation mechanism is crucial before sending each subtask $S_t = S_j^i$ at the time step $t$ to the Actor module. To address this, we designed a new module called Step-Check. Through extensive investigation, we discovered that during GUI task testing, perfect execution plans are rarely feasible due to the unpredictable nature of real application environments. Most software retains user preferences (e.g., remember the last configuration of user), meaning that when executing a specific task, the plan $p$ generated by the Planner might not align with the actual state of the software. Therefore, the model must determine whether to proceed with a subtask $S_t$ based on the current state (screenshot: $V_t$, metadata: $M_t$).

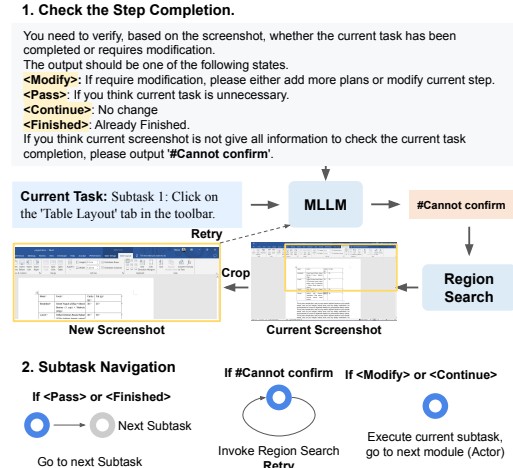

Figure 5: **Step-Check.** This module first checks the step completion status via an MLLM and then navigates to the current task processing.

As illustrated in Figure 5, we employ an MLLM to determine whether the current task has been completed or requires modification. We systematically categorize the possible outcomes into:

(1) <Modify>: Indicates that the subtask should be modified or additional subtasks should be added.

(2) <Pass>: Indicates that the current subtask is unnecessary and can be skipped.

(3) <Continue>: Indicates that the subtask is valid and should be executed as planned.

(4) <Finished>: Indicates that the subtask has already been completed and requires no further action.

In cases where the screenshot does not provide sufficient visual information for the MLLM to determine the output, the model outputs **"#Cannot confirm"**. When this occurs, we design a **Region Search** module implemented by an LLM. This module takes the corresponding GUI information extracted by the GUI parser and the task description of the current subtask to identify the relevant region. It then crops the region using the generated bounding box as the center coordinate, with the maximum width and height set to half of the original screenshot dimensions (ensure the region is smaller than the original screenshot). The cropped screenshot is subsequently sent to the Step-Check module to regenerate the decision.

Table 3: Success rate (%) of different agents on WorldGUI. Human* denotes the average performance of four expert participants who have watched the instructional video only once, similar to the model. Meta represents the meta task, while Aug. represents the augmented task.

| Method | Office | | Win. Usage | | Web | | Coding | | Media | | Overall |
|---|---|---|---|---|---|---|---|---|---|---|---|
| | Meta | Aug. | Meta | Aug. | Meta | Aug. | Meta | Aug. | Meta | Aug. | |
| Plan-Act w/ Gemini-2.0 | 8.9 | 3.2 | 8.3 | 3.4 | 28.6 | 16.2 | 18.2 | 2.2 | 10.0 | 2.0 | 6.9 |
| Plan-Act w/ GPT-4o | 13.3 | 10.1 | 8.3 | 2.3 | 23.8 | 11.1 | 9.1 | 2.2 | 10.0 | 2.0 | 8.5 |
| AssistGUI w/ GPT-4o | 26.7 | 16.1 | 29.2 | 7.9 | 33.3 | 20.2 | 27.3 | 11.1 | 10.0 | 8.2 | 16.5 |
| CCU w/ Claude-3.5-Sonnet | 28.9 | 19.3 | 29.2 | 14.6 | 71.4 | 32.3 | **54.6** | 22.2 | 30.0 | 6.1 | 23.6 |
| UI-TARS-1.5 | 28.9 | 16.1 | 12.5 | 2.2 | 28.6 | 9.1 | 36.7 | 6.7 | 0.0 | 0.0 | 12.3 |
| Agent S2 | 33.3 | 16.5 | **70.8** | **59.6** | 52.4 | 45.5 | 45.5 | **37.8** | 20.0 | 16.3 | 34.2 |
| WorldGUI-Agent (Ours) | | | | | | | | | | | |
| w / Gemini-2.0 | 31.1 | 17.0 | 20.8 | 9.0 | 38.1 | 29.3 | 36.4 | 11.1 | 20.0 | 10.2 | 19.1 |
| w / GPT-4o | 42.2 | 24.3 | 41.7 | 11.2 | 47.6 | 35.4 | 45.5 | 15.6 | 40.0 | 12.2 | 26.0 |
| w / Claude-3.5-Sonnet | **57.8** | **32.6** | 50.0 | 19.1 | **76.2** | **46.5** | **54.6** | 26.7 | **50.0** | **18.4** | **36.0** |
| **Human*** | 88.9 | 83.5 | 100.0 | 89.9 | 95.2 | 80.8 | 81.8 | 77.8 | 90.0 | 85.7 | 85.3 |

## 3.4 ACTOR

The goal of the Actor is to translate natural language subtask $S_t$ into executable code $C_t$. Using an MLLM as the backbone model, the Actor processes metadata $m_t$ and screenshot $V_t$ as GUI context to generate precise executable actions, such as `click(100, 200)`. Additionally, it leverages the history of previous actions as memory to aid in generating subsequent actions. The generated actions will be executed in the environment, and then the new screenshot $V_{t+1}$ and metadata $m_{t+1}$ will be captured for the next processing.

## 3.5 ACTOR-CRITIC

**Post-Action Evaluation.** After generating an action, the Actor-Critic module evaluates subtask $S_{t-1}$ completion and makes corrections if necessary. As illustrated in Figure 6, in the first step, the module implemented by an MLLM compares screenshots $V_{t-1}$ (before action execution) and $V_t$ (after execution) while processing each subtask $S_t$ to determine the action correctness. The model outputs a `<Success>` flag to indicate task completion. If the `<Success>` flag is true, the current state $s_t$ = `<Next>`. If the `<Success>` flag is false (set $s_t$ = `<Critic>`) and the number of trial steps is below the maximum limit, the Actor-Critic module activates the **Locate GUI Elements** and **Actor Correction** processes. We introduce the module **Locate GUI Elements** to identify the relevant GUI elements and regenerate actions using the **Actor Correction** module. The corrected actions are then executed in the environment, generating updated observations ($\mathcal{O}_t$) that include new screenshots and metadata for the continued Actor-Critic iteration. The process repeats until

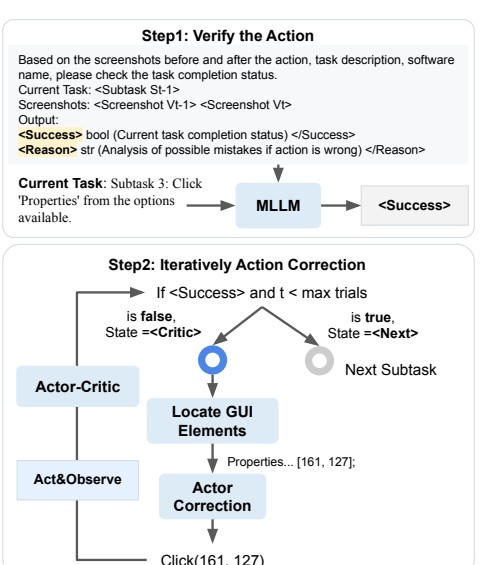

Figure 6: **Actor-Critic.** This module includes two parts: task verification and task correction. The design follows the *verify-then-correct* mechanism.

the `<Success>` flag is true or the maximum number of trials is reached.

## 4 EXPERIMENTAL RESULTS

**Implementation Details.** We implement the MLLM in our WorldGUI-Agent by using GPT-4o (OpenAI, 2023) (gpt-4o-2024-08-06) by default. For the computer mouse and keyboard control, we use the Python library PyAutoGUI. Following the AssistGUI (Gao et al., 2024), we use the GUI parser to obtain the position information of elements, e.g., buttons, icons, and text. We use some vision foundation models, such as Google OCR, to extract the text. By default, we use the center coordinates

Table 4: Success rate (%) of our WorldGUI-Agent with the ablation of different critical modules.

| Method | Office | | Win. Usage | | Web | | Coding | | Media | | Overall |
|---|---|---|---|---|---|---|---|---|---|---|---|
| | Meta | Aug. | Meta | Aug. | Meta | Aug. | Meta | Aug. | Meta | Aug. | |
| Full Model | 42.2 | 24.3 | 41.7 | 11.2 | 47.6 | 35.4 | 45.5 | 15.6 | 40.0 | 12.2 | 26.0 |
| – w/o Planner-Critic | 31.1 | 17.0 | 20.8 | 9.0 | 38.1 | 25.3 | 36.4 | 11.1 | 20.0 | 10.2 | 18.5 |
| – w/o Step-Check | 31.1 | 19.3 | 20.8 | 9.0 | 33.3 | 28.3 | 45.5 | 13.3 | 20.0 | 8.2 | 19.8 |
| – w/o Actor-Critic | 15.6 | 7.8 | 4.2 | 3.4 | 28.6 | 17.2 | 0.0 | 8.9 | 10.0 | 6.1 | 9.7 |

Table 5: Success rate (%) of our WorldGUI-Agent with the ablation of Instructional Video.

| Method | PPT | Word | Excel | Acrobat | VSCode | Overall |
|---|---|---|---|---|---|---|
| Full Model | 45.5 | 36.4 | 50.0 | 36.4 | 45.5 | 42.9 |
| w/o Inst. Video | 45.5 | 27.3 | 25.3 | 18.2 | 27.3 | 28.6 |

to represent the location of each element. All the testing is under the same screenshot resolution ($1920 \times 1080$). In all experiments, we set the max trials of the Actor-Critic to 3 for light interaction costs. For the total trials of each task, we set it to $4 \times N + 1$, where $N$ is set empirically.

**Evaluation.** Given that our WorldGUI includes 611 GUI tasks, we engaged four participants with strong coding and software backgrounds to test all tasks and document their evaluation results. **Metric.** Following the previous works of OSworld and AssistGUI, we use Success Rate (SR) as the metric.

**Baselines.** We implement the baseline approach called **Plan-Act** with different MLLMs as the base model. It focuses on investigating the basic capabilities of task planning and action prediction. Additionally, we compare our WorldGUI-Agent with two agentic frameworks and two SOTA GUI models: **AssistGUI** (Gao et al., 2024), **Agent-S2** (Agashe et al., 2025b), **Computer Use (Claud-3.5-Sonnet)** (Anthropic, 2024), and **UI-TARS-1.5** (Qin et al., 2025). AssistGUI and Agent-S2 are two prominent agentic frameworks designed for Desktop GUI Automation, which can plan the task and then execute the task step by step by following the query. We increase the base model to GPT-4o for AssistGUI and Claude-Sonnet-4 of Agent-S2 for fair performance. **Claude Computer Use (CCU)** is the leading proprietary model specially designed for computer use. We use the open-source implementation OOTB (Hu et al., 2024) as the codebase and then add the subtitle of instructional videos into the input prompt for a fair comparison. We also implement our **WorldGUI-Agent** with three different MLLMs to illustrate the effectiveness of our proposed universal agent framework.

## 4.1 MAIN RESULTS ON WORLDGUI

Table 3 reports the success rates (SR) of different agents and human experts on our WorldGUI benchmark, broken down by task type (Meta vs. Aug.) across five categories: Office, Win. Usage, Web, Coding, and Media. From these results we draw the following main conclusions.

**A large gap remains between agents and humans.** The best-performing agent (WorldGUI-Agent with Claude-3.5-Sonnet) achieves an overall SR of only 36.0%, which is less than half of the 85.3% attained by human experts. This stark contrast underscores the difficulty of our tasks and the need for further advances in desktop GUI automation.

**Agents generalize poorly to augmented tasks.** Across all methods, performance on Augmentation tasks (which introduce interface or context variations) is substantially lower than on their corresponding Meta tasks. For example, Claude-3.5-Sonnet in the Win. Usage category attains 50.0% on Meta tasks but drops to just 19.1% on Aug. tasks. This highlights the importance of dynamic testing to capture realistic human–computer interaction.

**Desktop applications pose a greater challenge than web tasks.** Every agent scores higher on Web tasks than on desktop application tasks. WorldGUI-Agent with Claude-3.5-Sonnet, for instance, jumps from 76.2% on Web Meta to only 57.8% on Office Meta, and the gap widens on their Augmentation counterparts. Thus, desktop GUI automation remain a frontier for computer use research.

**WorldGUI-Agent consistently outperforms a naive Plan-Act baseline.** By incorporating our three critical modules into the planning and execution loop, WorldGUI-Agent substantially improves success rates over the basic Plan-Act approach. Relative to Plan-Act, WorldGUI-Agent raises overall SR by +12.2% with Gemini-2.0, +17.5% with GPT-4o, and +12.4% with Claude-3.5-Sonnet, demonstrating the effectiveness of our design across multiple MLLMs.

Table 6: Experimental results on WindowsAgentArena (Bonatti et al., 2024). The reported results are from the (Bonatti et al., 2024) and (Agashe et al., 2025b).

| Method | Office | Web | Win. System | Coding | Media | Win. Utils | Overall |
|---|---|---|---|---|---|---|---|
| Phi3-V (Bonatti et al., 2024) | 0.0 | 6.9 | 8.3 | 0.0 | 6.2 | 0.0 | 3.5 |
| GPT-4o-mini (Bonatti et al., 2024) | 0.0 | 14.9 | 8.3 | 0.0 | 0.0 | 0.0 | 4.2 |
| GPT-4o (Bonatti et al., 2024) | 0.0 | 13.7 | 29.2 | 0.0 | 10.3 | 0.0 | 8.6 |
| NAVI (Bonatti et al., 2024) | 0.0 | 27.3 | 33.3 | 27.3 | **30.3** | 8.3 | 19.5 |
| Agent S (Agashe et al., 2025a) w/ GPT-4o | 0.0 | 13.3 | 45.8 | 29.2 | 19.1 | 22.2 | 18.2 |
| Agent S2 (Agashe et al., 2025b) w/ Claude-3.7-Sonnet | 7.0 | 16.4 | 54.2 | **62.5** | 28.6 | 33.3 | 29.8 |
| **WorldGUI-Agent w/ Claude-3.5-Sonnet** | **7.0** | **53.3** | **45.8** | 33.3 | 28.6 | **33.3** | **31.2** |

## 4.2 ABLATION STUDY

**Impact of different critical modules.** Table 4 presents the results of an ablation study on the three core components of WorldGUI-Agent across five application categories (Office, Windows Usage, Web, Coding, Media). The full model achieves an overall success rate (SR) of 26.0%. The effects of removing each component are as follows: **Planner-Critic:** Eliminating this module reduces overall SR to 18.5% (–7.5%), with substantial drops in Office (42.2% → 31.1%) and Web (47.6% → 38.1%) tasks, indicating its importance for refining initial plans. **Step-Check:** Without step-wise verification, SR decreases to 19.8% (–6.2%). The relatively smaller decline on Coding and Win. Usage tasks suggest that Step-Check excels at intercepting and correcting multi-step interaction errors. **Actor-Critic:** Removing the action-level critic causes SR to collapse to 9.7% (–16.3%). Performance on Coding Meta drops to 0.0% and Windows Usage Meta to 4.2%, highlighting the critical role of reward-driven action correction for action-level GUI operations. These results confirm that Planner-Critic, Step-Check, and Actor-Critic each contribute complementary benefits—plan refinement, intermediate validation, and action optimization—that are essential for the robustness and overall effectiveness of WorldGUI-Agent.

**Impact of Instructional Video.** In Table 5, we study the impact of removing the instructional video by modifying the prompt to include only the user query for generating the initial plan. In the Excel applications, we observe a significant performance decline, as their tasks are complex and difficult, and rely more heavily on additional domain knowledge for successful planning. In contrast, the MLLM performs relatively well on Win. Usage tasks, such as Settings and File Management, are where it has more inherent familiarity. These findings underscore the necessity of instructional videos for complex tasks like visual effect design, mirroring how users learning to build a slide often rely on tutorial videos.

## 4.3 RESULTS ON WINDOWSAGENTARENA BENCHMARK

Table 6 compares WorldGUI-Agent against four leading agents on the WindowsAgentArena benchmark. WorldGUI-Agent achieves a 31.2% overall SR, far surpassing GPT-4V (19.5%), GPT-4o (8.6%), GPT-4o-mini (4.2%), and Phi3-V (3.5%). Its gains are most pronounced in desktop categories: Office tasks (7.0% vs. 0%), Windows System (45.8% vs. 33.3%), and Windows Utilities (33.3% vs. 8.3%). On web browser, it reaches 53.3%, nearly double GPT-4V's 27.3%, and on coding tasks, it records 33.3% versus 27.3%. In media tasks, WorldGUI-Agent posts 28.6%, closely matching GPT-4V's 30.3%. These results underscore the necessity of integrated planning critique, step-check verification, and action-level feedback. These results demonstrate that our framework robustly handles both desktop GUI tasks and dynamic web environments, highlighting its versatility for real-world GUI automation.

## CONCLUSION

In this paper, we take the first step toward comprehensive GUI agent evaluation by introducing WorldGUI. In addition to the standard static testing processes, we incorporate dynamic testing procedures to ensure that WorldGUI effectively captures the complexity and dynamism of real-world GUI environments. Furthermore, we propose a universal agent framework, WorldGUI-Agent, built upon the critical thinking principle. This framework enables the agent to dynamically identify uncommon states and adjust its plans or actions accordingly. Finally, we evaluate WorldGUI-Agent powered by Claude-3.5-Sonnet on WorldGUI and WindowsAgentArena benchmarks, demonstrating the effectiveness across a variety of GUI tasks.

LIMITATION AND IMPACTS

In the current implementation of our agent, WorldGUI-Agent, more external tools have not been integrated into the GUI planning and action prediction processes to prioritize computational efficiency. Incorporating tools such as web search or file search into the agent's design could be a valuable future direction to improve performance. Additionally, due to the usage of the GUI Parser, which would increase the time costs because of the response speed of experimental desktop computers, it is still a tradeoff between performance and running time in the current GUI domain. We consider that if the base MLLM model is specifically trained with stronger planning and grounding ability, the running time would be sped up. It is noted that our agent framework is capable of working with any MLLM.

WorldGUI takes the first step of pushing the GUI automation into the dynamic testing process, as we found that real-world human-computer interactions are dynamic and unpredictable; existing GUI benchmarks fail to capture such dynamics to closely reflect the interactions. Our WorldGUI-Agent is a straightforward and universal agent framework by considering incorporates three critical modules to adaptively align the plan and actions with exact environment situations, which would be a good baseline for future agent development. For instance, incorporating more tools such as web search or file search into the planning module or action prediction module to realize more challenging tasks.

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

## A   ETHICS STATEMENT

This work introduces a benchmark for GUI interaction without involving any sensitive personal data or human subject experiments. All data are derived from publicly available or synthetically generated instructions, ensuring compliance with privacy and legal considerations. We acknowledge potential risks of misuse (e.g., surveillance), and therefore release the benchmark with clear documentation and intended use guidelines. We affirm adherence to the ICLR Code of Ethics throughout the research process.

## B   REPRODUCIBILITY STATEMENT

We have made extensive efforts to ensure reproducibility. Detailed dataset construction steps, task definitions, and evaluation protocols are described in Section 2 and the Appendix. Implementation details of experiments, including hyperparameters and evaluation scripts, are provided in Section

4. In addition, we release the benchmark dataset and evaluation code as anonymous supplementary materials to enable independent verification of our results.

## C   LLM USAGE STATEMENT

We employed large language models (LLMs) as auxiliary tools during manuscript preparation. Their use was strictly limited to surface-level editing tasks, including grammar correction, minor rephrasing, and stylistic improvements to enhance readability. At no point did we rely on LLMs for generating research ideas, methods, experiments, or conclusions. All technical content and analysis presented in this paper are the sole work of the authors.

## D   RELATED WORK

### D.1   GUI BENCHMARKS

GUI benchmarks are essential for evaluating the performance and robustness of GUI agents. For web applications, WebShop (Yao et al., 2022), WebArena (Zhou et al.), and WebVoyager (He et al., 2024) focus on creating the GUI tasks in a web browsing scenario. In OS environments, OSWorld (Xie et al., 2024) is a comprehensive benchmark, including various operating systems with real applications. Mobile benchmarks such as MobileAgent (Wang et al., 2024) and AppAgent (Zhang et al., 2023) propose two GUI benchmarks of mobile applications. Windows-related benchmarks like AssistGUI (Gao et al., 2024) and WindowAgentArena (Bonatti et al., 2024) propose a list of real tasks in the Windows platform. However, these online testing GUI benchmarks primarily rely on a static testing process and do not adequately capture the complexity and dynamic nature of GUI environments. As a result, they are insufficient for comprehensively evaluating GUI agents.

### D.2   GUI AGENTS

CogAgent (Hong et al., 2024) is a vision language model focused on GUI understanding to facilitate GUI navigation, while SeeClick (Cheng et al., 2024) and SeeAct (Zheng et al., 2024) focus on the GUI grounding for enhancing the task performance. MobileAgent (Wang et al., 2024) and AppAgent (Zhang et al., 2023) are proposed to design the agent on the mobile device. Ferret-UI (You et al., 2025) is another representative work focusing on enhancing the grounding ability in the IOS platform. These agents have shown their ability in GUI understanding (e.g., GUI elements grounding) or action prediction, but still face limitations in handling dynamic and complicated full GUI tasks. Therefore, to enhance GUI automation in dynamic environments, we propose WorldGUI-Agent, which improves adaptability in complex GUI settings and enables agents to effectively handle unpredictable interface changes. The components comparison of our WorldGUI-Agent and other closely related agents is shown in Table 7.

Table 7: Comparison with other closely related agents. Most existing agents solely focus on post-action evaluation but omit the post-planning critique and pre-action validation in handling dynamic GUI environments.

| Method | Post-Planning Critique | Pre-Action Validation | Post-Action Evaluation |
|---|---|---|---|
| Mobile-Agent (Wang et al., 2024) | ✗ | ✗ | ✓ |
| Mobile-Agent-V2 (Wang et al., 2024) | ✗ | ✗ | ✓ |
| AssistGUI (Gao et al., 2024) | ✗ | ✗ | ✓ |
| Agent-S (Agashe et al., 2025a) | ✗ | ✗ | ✓ |
| Mobile-Agent-E (Wang et al., 2025) | ✗ | ✗ | ✓ |
| WorldGUI-Agent (ours) | ✓ | ✓ | ✓ |

### D.3   CRITICAL THINKING IN AGENTS

Recent advancements in foundation models and agents, particularly in LLMs such as OpenAI-o1 (OpenAI, 2024) and Deepseek-R1 (DeepSeek-AI et al., 2025), have increasingly incorporated thinking processes before providing answers to effectively handle challenging reasoning tasks. The LLM-based agents utilize *verify-then-correct* process to evaluate and refine intermediate reasoning steps

Table 8: Performance comparison between **WorldGUI-Agent** (with Claude-3.5-Sonnet and Claude-Sonnet-4) and **Agent-S2** (Claude-Sonnet-4). Results are reported across five representative applications.

| Method | PPT | | VSCode | | Acrobat | | VLC | | File Explorer | |
|---|---|---|---|---|---|---|---|---|---|---|
| | Meta | Aug. | Meta | Aug. | Meta | Aug. | Meta | Aug. | Meta | Aug. |
| Agent-S2 w/ Claude-Sonnet-4 | 45.5 | 18.9 | 45.5 | 37.8 | 18.2 | 14.5 | 20.0 | 16.3 | 60.0 | 64.7 |
| WorldGUI-Agent w/ Claude-3.5-Sonnet | 54.5 | 39.6 | 54.5 | 26.7 | 63.6 | 20.0 | 50.0 | 18.4 | 50.0 | 17.6 |
| WorldGUI-Agent w/ Claude-Sonnet-4 | 63.6 | 52.8 | 54.5 | 28.9 | 54.5 | 30.9 | 40.0 | 28.6 | 70.0 | 26.5 |

Table 9: Performance comparison between **WorldGUI-Agent** (with UI-TARS-1.5) and **UI-TARS-1.5**.

| Method | PPT | | Acrobat | |
|---|---|---|---|---|
| | Meta | Aug. | Meta | Aug. |
| WorldGUI-Agent w/ UI-TARS-1.5 | 36.6 | 18.9 | 36.4 | 9.1 |
| UI-TARS-1.5 | 27.3 | 17.0 | 9.1 | 1.8 |

or outputs, ensuring logical coherence and consistency. One notable LLM-based agent framework, Reflexion (Shinn et al., 2024), demonstrates the effectiveness of self-reflection in solving complex tasks. Furthermore, CRITIC (Gou et al., 2023) integrates external tools into the critique process, leveraging them to improve performance. Noticing that the GUI task is lengthy and complicated, the *verify-then-correct* process is highly suitable for the GUI scenario. Which is not only aims to enhance the reasoning performance but is also indispensable to designing the key module Actor-Critic (Konda & Tsitsiklis, 1999) to ensure task completion. A closely related work, AssistGUI (Gao et al., 2024), integrates a critical module only after the Actor module to evaluate action completion. Building upon it, we introduce two additional critical modules: Planner-Critic, applied after the Planner, and Step-Check, applied before the Actor. These two modules lead to a universal and fundamental GUI agent framework **WorldGUI-Agent** which will provide insights for future GUI agent design.

## E  ADDITIONAL EXPERIMENTS

As shown in Table 8, Agent-S2 (Agashe et al., 2025b) shows competitive results as compared with our WorldGUI-Agent. We also test on two representative office software to compare the effectiveness of our proposed agentic framework by replacing the base model with UI-TARS-1.5 (Qin et al., 2025) in Table 9. It is noted that to improve the performance of UI-TARS-1.5, we use the GPT-4o to task planning, as we found that UI-TARS struggles with understanding complex desktop software layout and cannot capture the dynamic initial condition changes. We use GPT-4o for better implementation.

## F  DATA

### F.1  ANNOTATORS

In this work, we have four annotators: A, B, C, and D. The team comprises one PhD student, one Master's student, and two undergraduate students. Prior to annotation, all annotators receive training on using the applications in WorldGUI to ensure high-quality annotations. For the 10 desktop applications, we divide the software into four parts, assigning each part to a different annotator. For the human tests presented in Table 3, the annotators demonstrate tasks on software that they did not annotate. As shown in Table 1, each annotator is responsible for different software during both the annotation and human testing phases to make the soundness of the Human Test results.

### F.2  CREATING AUGMENTED TASKS

In our study, to simulate dynamic testing processing in real GUI interactions, we propose to design GUI tasks with various initial tasks. Specifically, we propose pre-actions before executing the task. The pre-actions primarily serve two purposes: **1) Simulating Intermediate Task States**:

Table 10: The annotation arrangement during the annotation and human testing phases by different annotators.

| Annotators | Annotation Phase | Human Test Phase |
|---|---|---|
| A | PowerPoint, Word, Excel | VSCode, VLC Player, Web |
| B | Adobe Acrobat, VLC Player | Excel, Settings |
| C | Settings, Web | PowerPoint, File Explorer, Youtube |
| D | VSCode, File Explorer | Word, Adobe Acrobat |

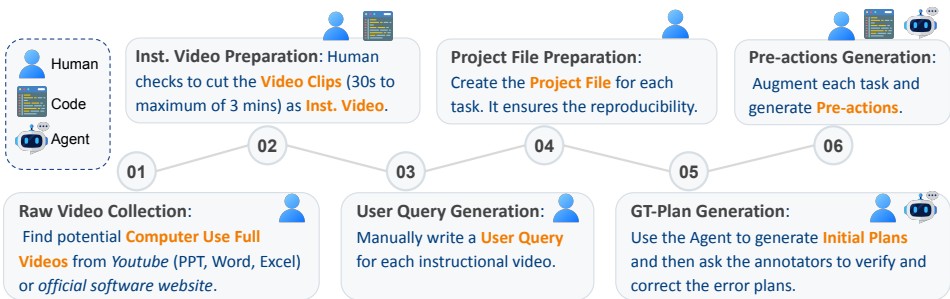

Figure 7: Pipeline of Data Construction. Human: Represents the annotators. Code: Refers to the scripts (e.g., Python Code) utilized to achieve the goal. Agent: We design an agent built upon the MLLMs to achieve the goal.

Pre-actions can complete specific steps of a task, creating a starting point from an intermediate state. This approach addresses scenarios where users may seek AI assistance because they are unable to complete a task. For example, if the task involves opening a dropdown menu, the pre-action may pre-open the menu. If the agent fails to recognize this precondition and follows its plan to click the menu again, it might inadvertently close the menu, causing task failure.

### F.3 INTRODUCING DIVERSE INITIAL CONTEXT STATES

Pre-actions can also introduce variations in the initial state, such as opening random tabs or settings. This ensures that the starting state is unconventional, challenging the agent to adapt by modifying its plan or adding new steps. We illustrate one example in Figure 8. Here, the meta task and augmented task, have the same user query and instructional video and it will ideally have the same final state. We additionally provide more examples about augmenting the meta task in Figure 9.

## G DATA STATISTICS

### G.1 TASK TAXONOMY AND LENGTHS

We present the detailed data statistics about WorldGUI in Figure 10 and Table 12. In total, we have 111 meta tasks across 10 widely-used desktop software to construct the data of WorldGUI. By augmenting the default task state 5 times, we obtain 500 augmented tasks. In summary, we have 611 tasks in WorldGUI for testing GUI agents. The task lengths are shown on the left of Figure 10. The average length of meta tasks is 6, and it would be increased by 1 due to the context variety. As the overall performance of WorldGUI is around 36%

### G.2 AUGMENTATION TASKS TYPE ANALYSIS

As we summarize, the real GUI scenarios include: (1) The software interface is not in its default state. (2) The human-computer interactions may start from the intermediate state of a specific task. We propose to create the augmented tasks for each meta task to simulate authentic GUI interactions. Our augmentations lie in two main groups: (1) simulating the intermediate states and (2) introducing diverse initial states. We divide the two groups into three exact types: Add-step, Trim-step, and

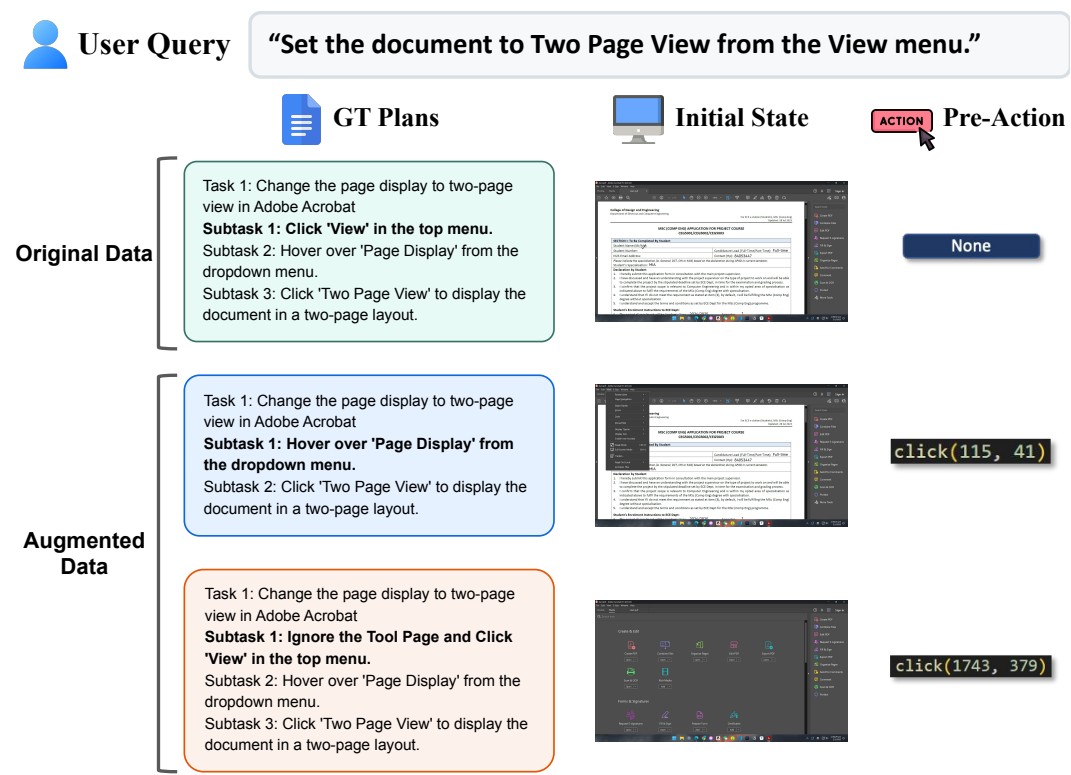

Figure 8: An example of augmenting one GUI task with manually aug the initial state and then using the execution scripts and corresponding agents to obtain the pre-action for each augmented case.

Adjust-step. For Add-step, it represents various unrelated state augmentations to simulate the scenario that we may start the agent-computer interactions in another unrelated task or interfaces, the agent should replan the task to add necessary steps. For Trim-step, it represents that we finish several steps of a long task and make the task in an intermediate state. For Adjust-step, it is usually a small modification of the existing state, such as changing the interface by clicking another Tab or clicking a button to open an unrelated dropdown menu. Most of the time, it would not require new steps to return to the target task progress. This augmentation may mislead the agent in state understanding, making them jump or miss the key steps. As shown in Figure 11, the manually created augmentations mainly belong to the add-step. Adjust-step could be the second-largest application, except for the File Explorer. Due to the low complexity of the interfaces of File Explorer, we cannot create many augmentations for adjust-step.

### G.3 TASK DIFFICULTY ANALYSIS

Figure 12 shows the distribution of the task difficulty across desktop applications. We annotated the task difficulty level based the subjective software usage experience. The results indicate that the tasks in Adobe Acrobat and VLC player are more challenging. The tasks in Excel, PowerPoint, and Word are more at the medium and simple levels. By considering the Success Rate and task length on these tasks, one can know that the tasks are easy for humans, but hard for current GUI agents. According to VSCode and File Explorer, and YouTube applications, the tasks are easier than in other applications. Overall, the task difficulty of created data is diverse across different applications, and there is still a need for stronger agents focusing on handling desktop-oriented GUI tasks.

The task details about the user query and the pre-actions are included in the metadata JSON file in the supplementary materials. The project file, instruction video, and augmentation files can be found in the provided data link.

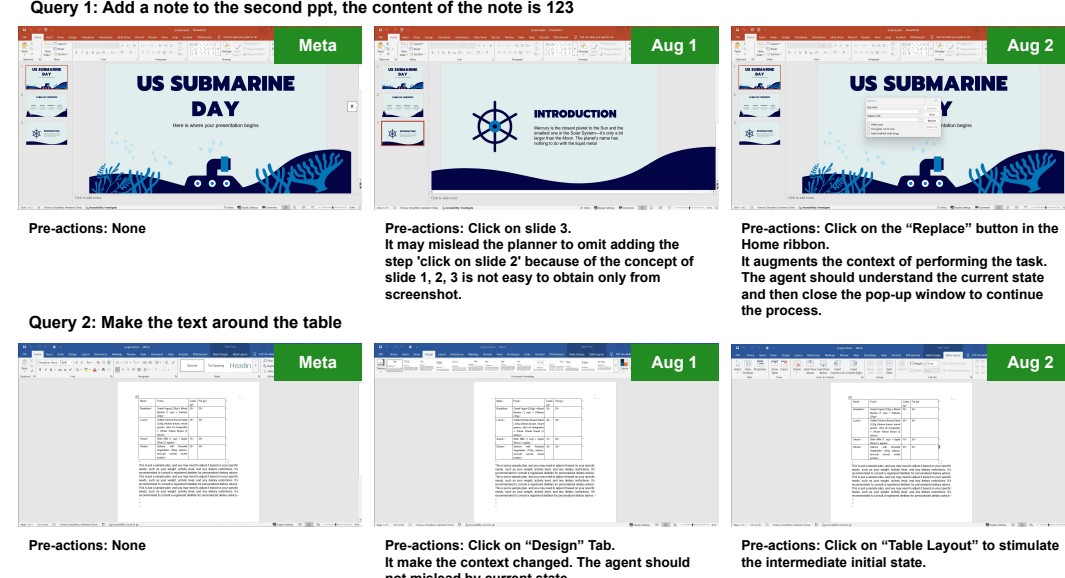

Figure 9: We present the examples of conducting the augmentations from the meta task.

Table 11: Task category, task activities, and project file of the desktop applications in WorldGUI.

| Category | Applications | All Task | Task Activities | Project File Type |
| --- | --- | --- | --- | --- |
| Office | PowerPoint | 64 | Change the content style and layout; Design new effects | project.pptx |
| Office | Word | 63 | Formatting the content style and layout | project.docx |
| Office | Excel | 70 | Table formatting; Data management and processing | project.xlsx |
| Office | Adobe Acrobat | 66 | Automatic add electric signature; Document management | project.pdf |
| Coding | VSCode | 56 | Code editing; Software configuration | vscode.exe |
| Windows Usage | Settings | 69 | Advanced personalized and safety settings; | ms-Settings |
| Windows Usage | File Explorer | 44 | File management: Add, delete, rename, and move files | explorer.exe |
| Web Usage | Web Browser | 59 | Web operation | web browser + URL |
| Web Usage | Youtube (Online) | 61 | Video and account configurations | web browser + URL |
| Media | VLC Player | 59 | Video editing and creation | project.mp4 |

## H  DETAILED EXPERIMENTAL RESULTS

Table 13 shows the detailed results of WorldGUI-Agent across individual applications in WindowsAgentArena (Bonatti et al., 2024) benchmark. The results of this related Windows-centric interactive GUI benchmark indicate that current the desktop GUI tasks are more challenging than web tasks. As we complete 11 out of 17 tasks in Web Browsing, a similar phenomenon is also discovered in Table 4.

## I  COMPUTATIONAL COSTS DISCUSSION

The average number of execution steps and tokens consumed are shown below Table 14. The execution steps are calculated based on our experimental log files, while the token costs are sampled from representative tasks in each category by taking Actor module as an example.

Take a Windows Setting task as an example, we provide detailed time costs across different modules tested on a desktop PC with AMD Ryzen 7 5800H CPU. Task length: 6 (generated by Planner+Planner-Critic). To facilitate a fair comparison, we additionally selected two of the latest SOTA agents, Agent-S (2024-10-08) and Agent-S2 (2025-04-01), and measured their run times on the same successful task under identical hardware and the same base MLLM (Claude-Sonnet-4). The results are shown in Tables 15, 16, 17. To summarize, our **WorldGUI-Agent** shows a competitive running time of 129.55s, as compared with Agent-S (131.98s) and Agent-S2 (108.64s). The main

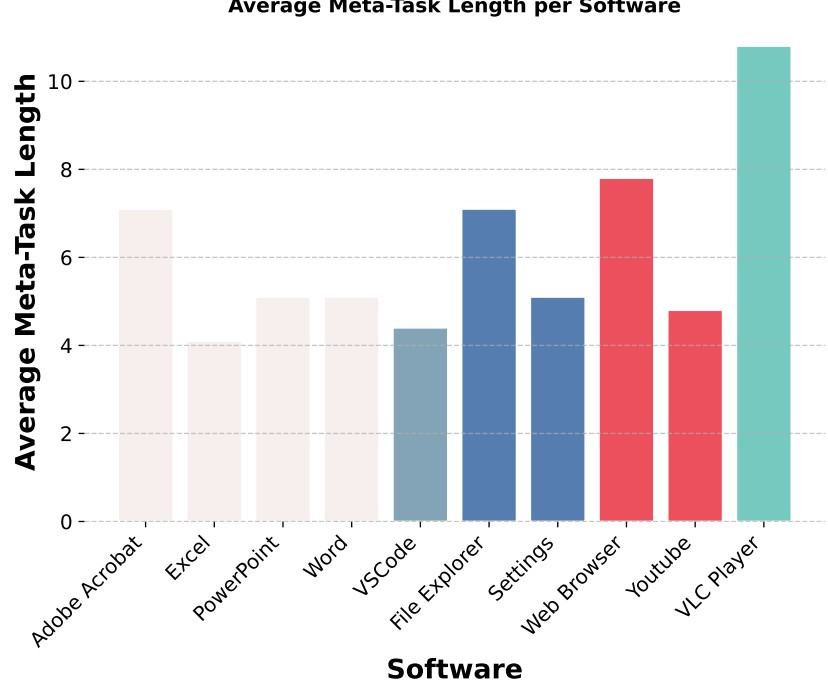

Figure 10: Distribution of Software taxonomy and the distribution of task length.

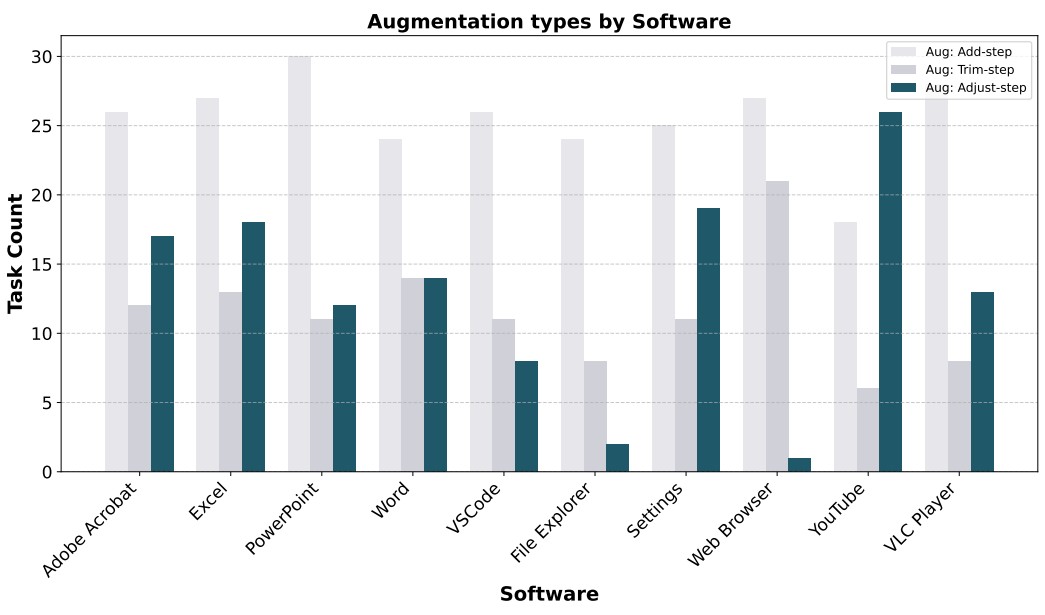

Figure 11: The distribution of different augmentation types.

computational costs of our designed modules are largely affected by the underlying large multimodal model, leaving room for acceleration optimization.

Since desktop GUI automation is still in its early stages, such computational costs are currently unavoidable. For reference, even OpenAI's Deep Research reportedly takes over 10 minutes in daily

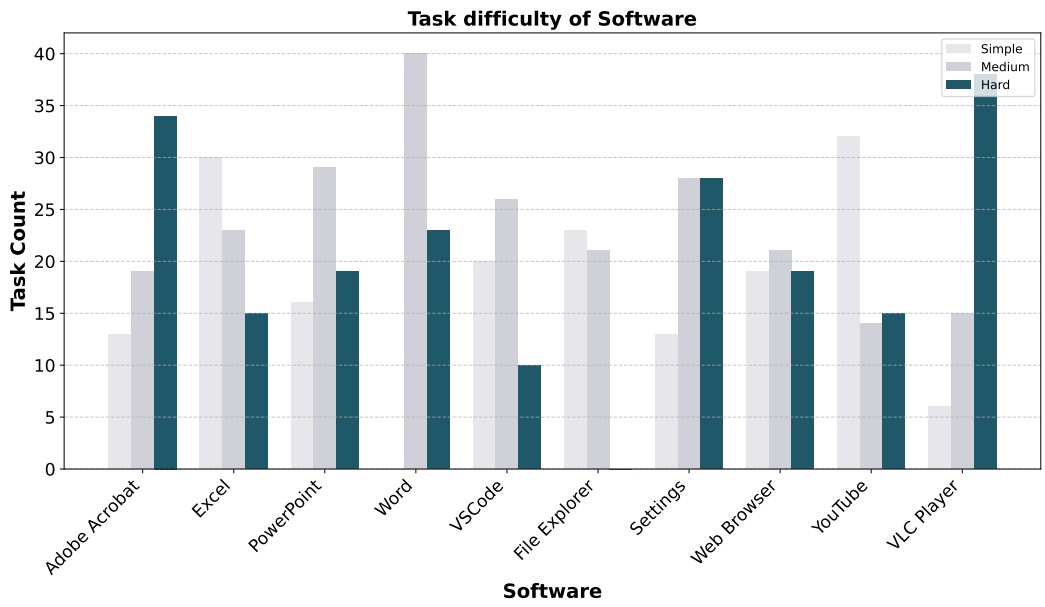

Figure 12: The distribution of different task difficulty.

Table 12: Task statistics of WorldGUI.

| Category | Software | Meta Count | Aug Count | Meta-Task Length (Avg.) |
|---|---|---|---|---|
| Office | Adobe Acrobat | 11 | 55 | 7.1 |
| Office | Excel | 12 | 58 | 4.1 |
| Office | PowerPoint | 11 | 53 | 5.1 |
| Office | Word | 11 | 52 | 5.1 |
| Coding | VSCode | 11 | 45 | 4.4 |
| Windows Usage | File Explorer | 10 | 34 | 7.1 |
| Windows Usage | Settings | 14 | 55 | 5.1 |
| Web Usage | Web Browser | 10 | 49 | 7.8 |
| Web Usage | YouTube | 11 | 50 | 4.8 |
| Media | VLC Player | 10 | 49 | 10.8 |
| Total (Average) | – | 111 | 500 | 6.0 |

usage. According to OpenAI Operator's report, achieving 38.1% on OS-World requires over 100 steps, which is similarly costly. In summary, there remains a clear tradeoff between performance and time costs in GUI automation, and this challenge is shared across the community.

## J  EXAMPLES OF AUGMENTATIONS

In this section, we present several augmentation examples in Figures 13, 14, 15, 16, 17, 18, 19. It is noted that our augmentations are not only making the first step changing but also require the agent add new step in its second step. For instance, in Figure 13, our augmentation is about click on Data tab in the ribbon, in the default software state, the Merge & Center button exhibit in the Home tab, there is no need to click on Home tab, after our augmentations, the agent should add a new task "Click on Home Tab" before it click on the Merge & Center button. Similarly, in Figure 18, the text editing buttons are under the Home Tab, if we augment the initial state with other Tab like Animation Tab, after the first step "Select the text 'US SUBMARINE DAY' ", the agent should add a new step like "Click on Home Tab" back to the default state for task execution. Except for adding new steps, we also present an example about adjust the step in Figure 14, as the target is about merging cells

Table 13: Detailed experimental results of WorldGUI-Agent across individual applications in WindowsAgentArena (Bonatti et al., 2024).

| Domain | Application | #Tasks | #Successes | SR (%) |
|--------|-------------|--------|------------|--------|
| Web Browsing | chrome | 17 | 11 | 64.71 |
| Windows Utilities | clock | 4 | 2 | 50.00 |
| Windows System | file_explorer | 19 | 7 | 36.84 |
| Office | libreoffice_calc | 24 | 1 | 4.17 |
| Office | libreoffice_writer | 19 | 2 | 10.53 |
| Windows Utilities | microsoft_paint | 3 | 1 | 33.33 |
| Web Browsing | msedge | 13 | 5 | 38.46 |
| Windows Utilities | notepad | 2 | 1 | 50.00 |
| Windows System | settings | 5 | 4 | 80.00 |
| Media & Video | vlc | 21 | 6 | 28.57 |
| Coding | vs_code | 24 | 8 | 33.33 |
| Windows Utilities | windows_calc | 3 | 0 | 0.00 |
| **Overall** | | **154** | **48** | **31.17** |

Table 14: Average execution steps and token costs on different software.

| Application category | Average execution steps | Input tokens per step (Actor) per task | Output tokens per step (Actor) per task |
|----------------------|-------------------------|----------------------------------------|------------------------------------------|
| Office | ∼23 | 2350 | 212 |
| Win. Usage | ∼20 | 1929 | 108 |
| Web | ∼17 | 1637 | 84 |

A1 to K1, we augment the initial state by selecting A2 to K2. Such a slight difference may mislead the agent to perceive such a minor difference, and the agent may jump the first step about selecting the correct cells lead to finally unsucess. In Figure 15 and Figure 19, we show two examples of introducing pop-up window in the initial state which require the agents accurately identify the pop-up windows and correctly close it by replanning the task based on the visual screenshot not only strictly planning based on inherited knowledge or the instructinal videos. In Figure 16, we show an example of changing the interface by clicking the Data tab to hide the Merge & Center button under the Home tab. In Figure 17, we complete the first step about selecting A1 to K1, which requires the agent to jump this step to reduce the time costs.

## K WORLDGUI-AGENT REASONING LOOP ALGORITHM

In this section, we provide the details of our reasoning loop algorithm in Algorithm 1.

## L QUALITATIVE RESULTS

(1) In Figure 20, we present a successful prediction example, demonstrating that the WorldGUI can effectively plan each step for a task, accurately perceive specific elements in the GUI, and convert them into the correct action code. Additionally, we display the parsed GUI elements, which can accurately identify most content, including small icons and dense text elements.

(2) We provide the visualization results of using Planner-Critic, Step-Check, and Actor-Critic in Figure 22, Figure 23, and Figure 24. These qualitative results demonstrate the effectiveness of these critical modules in GUI automation.

(3) We also highlight some common errors encountered. 1) The model has the difficulty of obtaining the desired information when we augment the task by invoking the dropdown menu of the Settings application. As shown in the left of Figure 25, when we click on the 'System' button on the left, it is challenging for our model to extract the button's position as it is hidden. Such cases require the model to have a higher level of ability to delete the content in the input box or click on the blank area. 2) As shown in the right of Figure 25, the model has difficulty dragging a bar to achieve the

Table 15: Running time with **WorldGUI-Agent (ours)**.

| Subtask Index | Executed Modules | Time (seconds) |
|---|---|---|
| 0 | Planner | 4.48 |
| 0 | Planner-Critic | 11.47 |
| 1 | Parser | 2.03 |
| 1 | Step-Check | 4.95 |
| 1 | Actor | 7.47 |
| 1 | Parser | 2.07 |
| 1 | Actor-Critic | 7.38 |
| 2 | Parser | 2.03 |
| 2 | Step-Check | 6.71 |
| 2 | Actor | 6.05 |
| 2 | Parser | 2.07 |
| 2 | Actor-Critic | 7.97 |
| 3 | Parser | 2.12 |
| 3 | Step-Check | 6.35 |
| 3 | Actor | 6.71 |
| 3 | Parser | 2.22 |
| 3 | Actor-Critic | 10.06 |
| 4 | Parser | 2.01 |
| 4 | Step-Check | 6.19 |
| 4 | Actor | 8.54 |
| 4 | Parser | 2.40 |
| 4 | Actor-Critic | 9.55 |
| 5 | Parser | 2.22 |
| 5 | Step-Check | 6.50 |
| Total | - | 129.55 |
| Average (per action) | - | 22.72 |

Table 16: Running time with **Agent-S**.

| Step | Executed Modules | Time (seconds) |
|---|---|---|
| 0 | Manager | 2.02 |
| 1 | Manager | 10.11 |
| 2 | Worker | 16.05 |
| 3 | Worker | 12.38 |
| 4 | Worker | 17.30 |
| 5 | Worker | 12.26 |
| 6 | Worker | 15.17 |
| 7 | Worker | 15.46 |
| 8 | Worker | 11.83 |
| 9 | Worker | 19.40 |
| Total | - | 131.98 |
| Average (per worker) | - | 14.98 |

Table 17: Running time with **Agent-S2**.

| Step | Executed Modules | Time (seconds) |
|------|------------------|----------------|
| 0 | Manager | 11.87 |
| 1 | Worker | 7.27 |
| 2 | Worker | 13.21 |
| 3 | Worker | 13.59 |
| 4 | Worker | 14.04 |
| 5 | Manager | 10.61 |
| 6 | Worker | 9.87 |
| 7 | Manager | 7.53 |
| 8 | Worker | 20.65 |
| Total | - | 108.64 |
| Average (per worker) | - | 13.22 |

**User Query: Merge A1:K1**

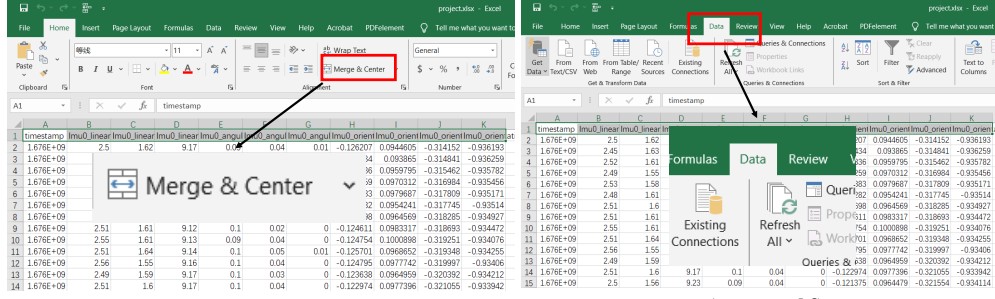

Default State         Augmented State

**GT Plan:**
Task 1: Select A1:K1
Subtask 1: Select the cells A1 through K1.
Task 2: Merge cells
Subtask 1: Click the **Merge & Center** command.
Subtask 2: Click 'OK' to confirm the change.

Figure 13: Augmented example of an Excel Task.

desired value. 3) The model struggles with the visual choice when there is no text information in the screenshot, as shown on the left of Figure 26. The subtask aims to select the center button, but the current model makes it hard to detect the center choice only from the screenshot. 4) The model cannot successfully locate the position of the input box, as the GUI parser will easily locate the text location 'Replace with', it always outputs the action like clicking on the 'Replace with', which will destroy the whole task success.

**User Query: Merge A1:K1**

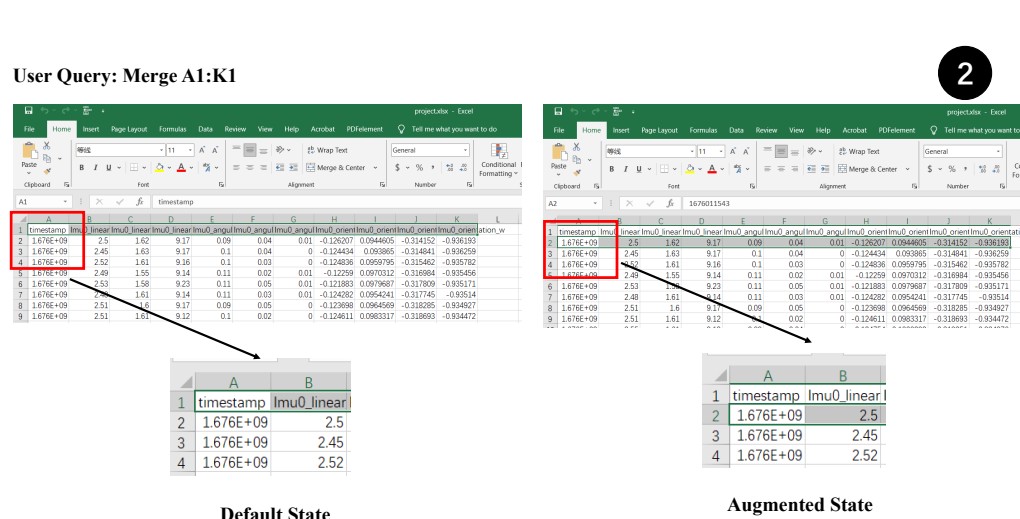

Default State

Augmented State

Figure 14: Augmented example of an Excel Task.

**User Query: Merge A1:K1**

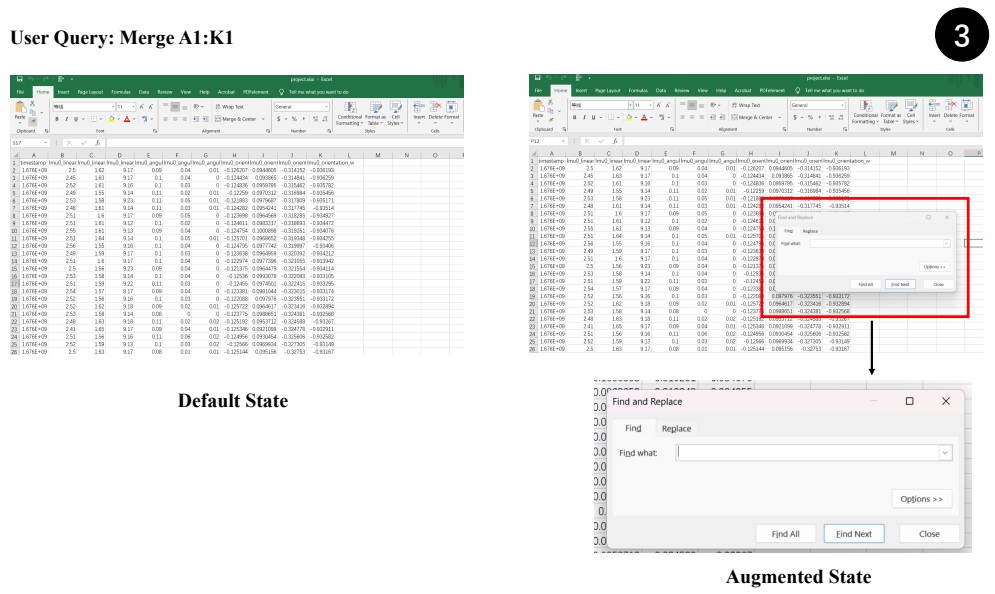

Default State

Augmented State

Figure 15: Augmented example of an Excel Task.

**User Query: Merge A1:K1**

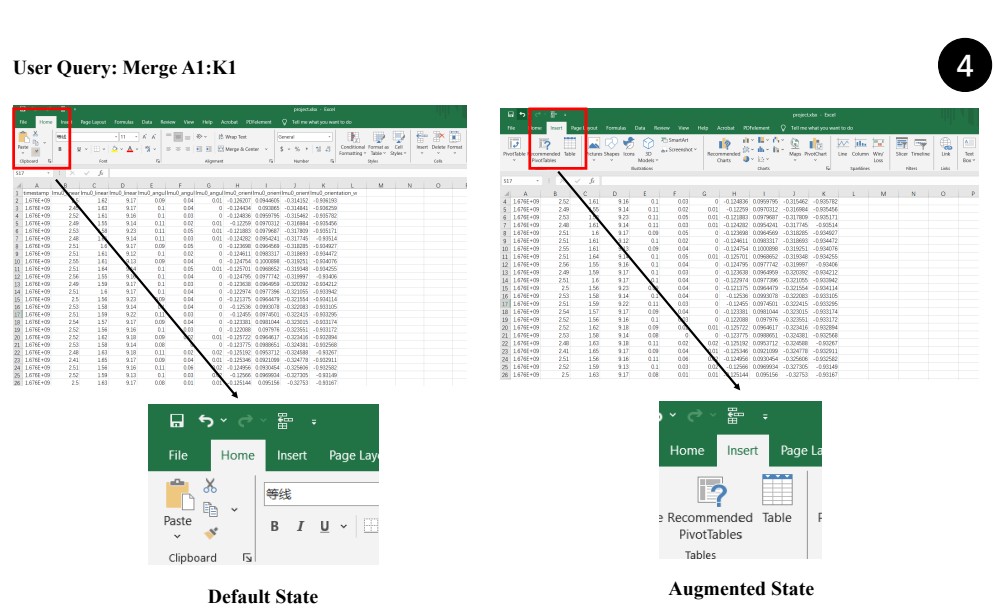

Default State            Augmented State

Figure 16: Augmented example of an Excel Task.

**User Query: Merge A1:K1**

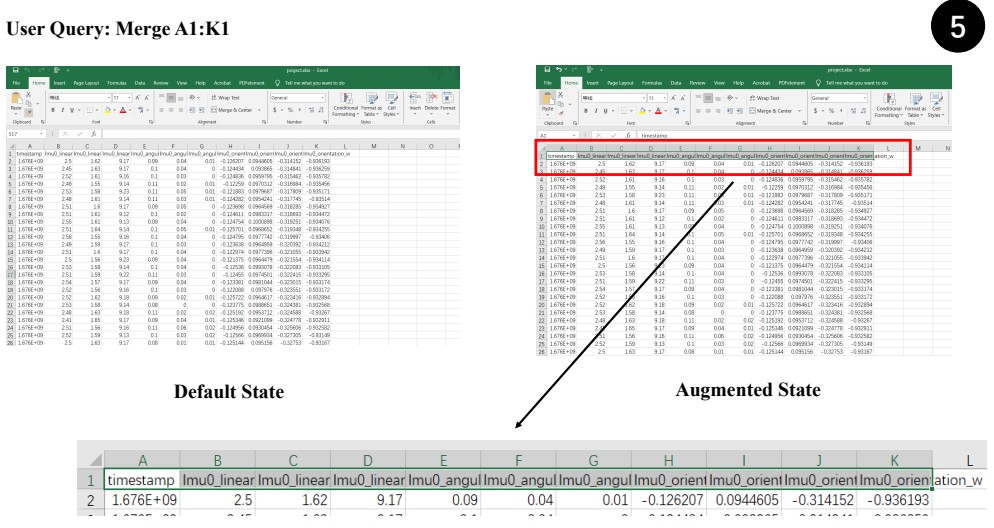

Default State            Augmented State

Figure 17: Augmented example of an Excel Task.

**User Query: Set US SUBMARINE DAY in the first ppt to fontsize 40**

**Default State**

**Augmented State**

Figure 18: Augmented example of a PowerPoint Task.

**User Query: Select all text and apply numbered list for them. Use '1, 2, 3' symbol of numbered list.**

**Default State**

**Augmented State**

Figure 19: Augmented example of a PowerPoint Task.

**Algorithm 1** WorldGUI-Agent Reasoning Loop Algorithm

**Input:** State $s$, Action Code $C$, Screenshot $V$, Metadata $m$, Current subtask $S$, Critic_count $z$

Generate task plan $p$ with **Planner** and **Planner-Critic**

Initial current subtask $S_{t=0} = S_1^1$, where $S_1^0$ is the 1-th subtask in the 1-th milestone of $p$.

Initial $s_0 = <Continue>$

**while** $S_t$ is not end and $t < $ max trials **do**

    Observe metadata $m_t$ and Screenshot $V_t$ from Env.

    Obtain state $s_t$ by running **Step-Check**.

    **if** $s_t = <Next>$ **then**

        Go to the next task $S_{t+1} = next(S_t)$

    **end if**

    Check potential modification of subtask $S_t$

    Obtain action code $C_t$ by running **Actor**; Execute the action code $C_t$ in the Env.; Observe metadata $m_t$ and Screenshot $V_t$ from Env.

    Set $C_t = $ None; $t = t + 1$; Set state $s_t = <Critic>$ (For each subtask, the first step is finished, then execute the actor-critic process)

    Observe metadata $m_t$ and Screenshot $V_t$ from Env.

    Running **Actor-Critic** and obtain the state $s_t$

    **if** $s_t = <Next>$ **then**

        Go to the next task $S_{t+1} = next(S_t)$.

    **end if**

    **while** $s_t = <Critic>$ and $z < $ max critique trials **do**

        Running **Actor-Critic** and obtain the state $s_t$ and corrected action code $C_t$

        **if** $s_t = <Next>$ **then**

            Go to the next task $S_{t+1} = next(S_t)$.

        **end if**

        Execute the action code $C_t$ in the Env.; Observe metadata $m_t$ and Screenshot $V_t$ from Env.

        Set $C_t = $ None; $z = z + 1$

    **end while**

    Go to the next task $S_{t+1} = next(S_t)$

    $t = t + 1$

**end while**

Subtask 1: Open the 'Settings' application on your PC.    Subtask 2: Click on 'System' in the left sidebar.    Subtask 3: Click on 'Notifications' from the available options.

Subtask 4: Scroll down to the 'Notifications from apps and other senders' section.    Subtask 5: Locate the 'Battery saver' item in the list.    Subtask 6: Toggle the switch next to 'Battery saver' to the 'Off' position to disable notifications.

Figure 20: We show one successful prediction of our WorldGUI-Agent.

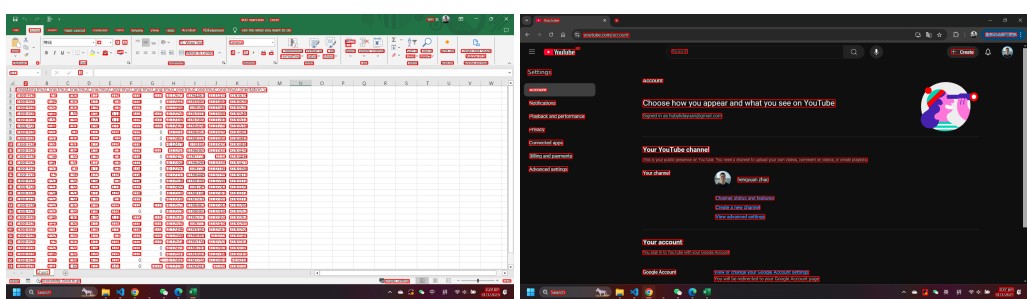

Figure 21: We show two examples of using GUI Parser to obtain the element position information.

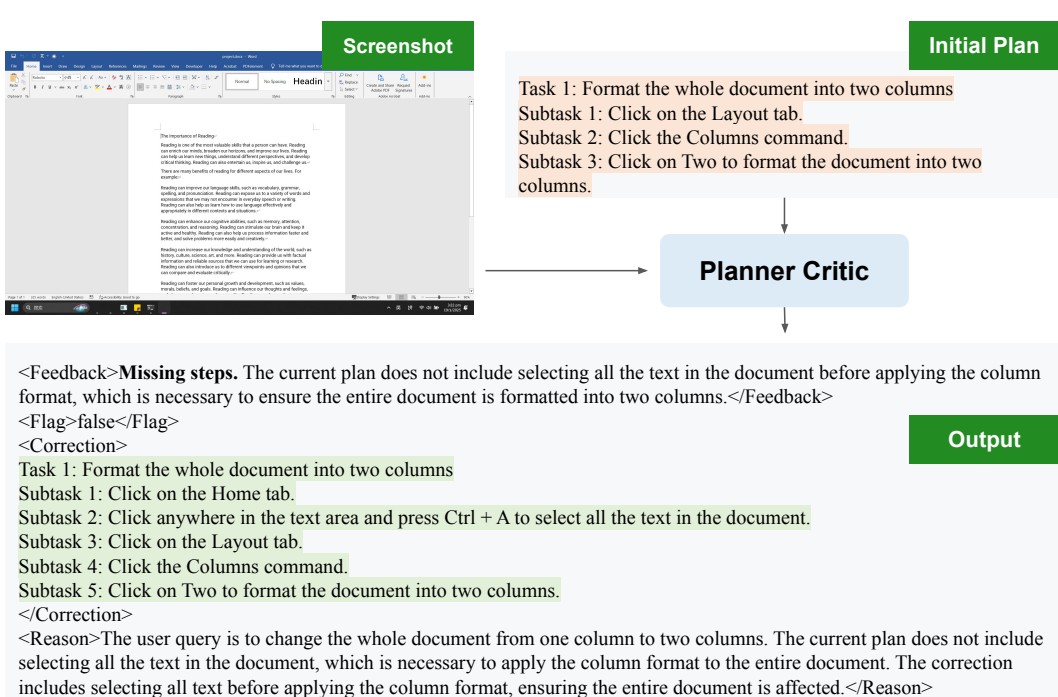

Figure 22: An example of using Planner-Critic to correct the plan.

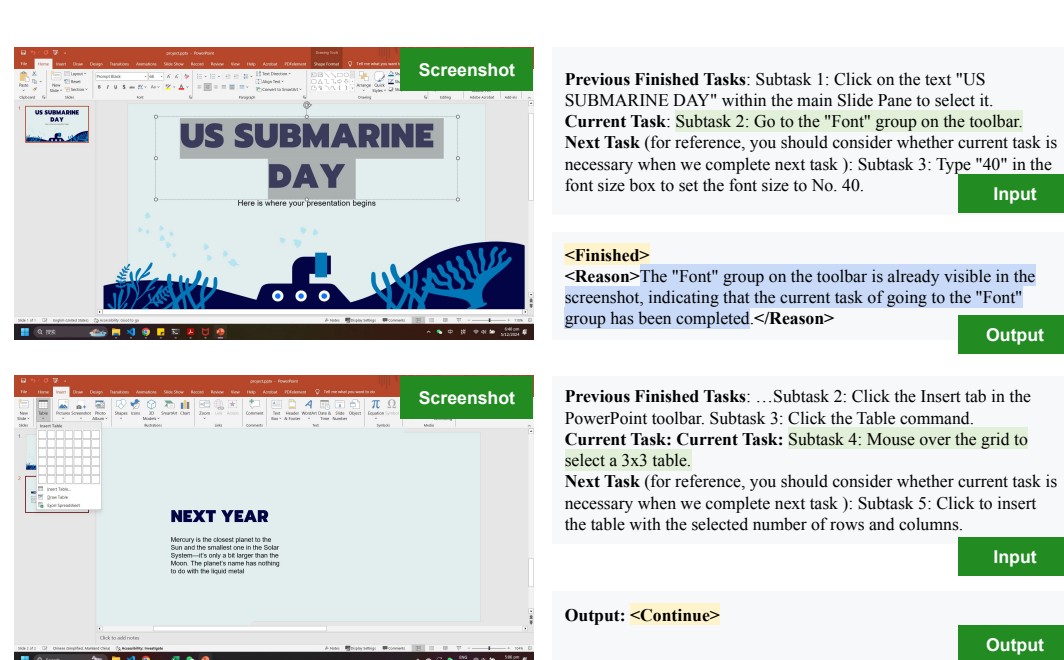

Figure 23: Two examples of using Step-Check to check the subtask status.

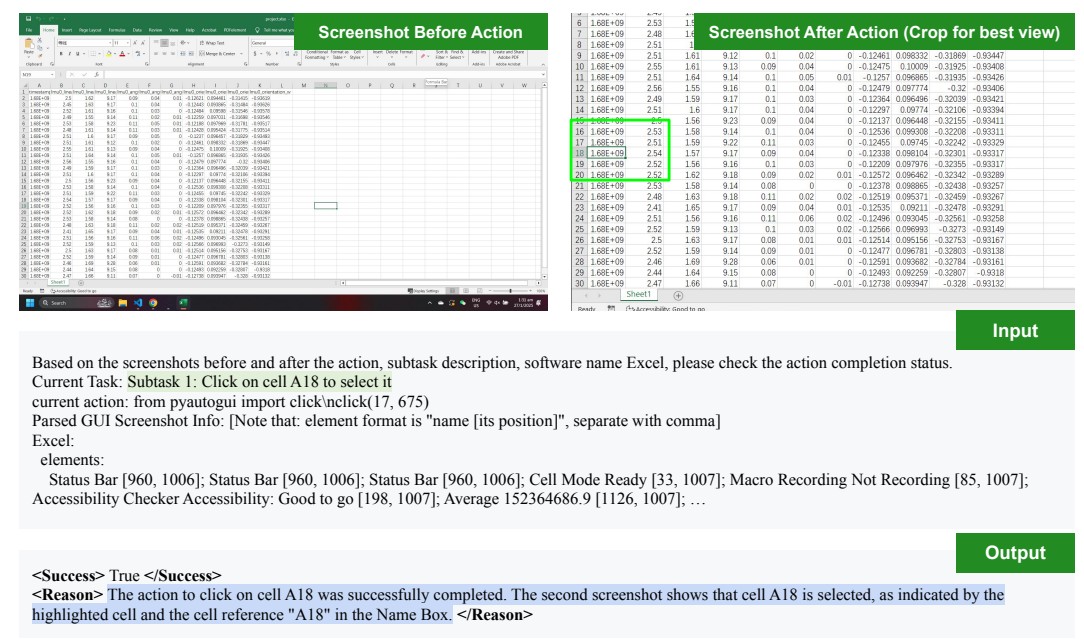

Figure 24: An example of using Actor-Critic to correct the actions.

**Query 1: Turn on Storage Sense**

**Subtask: Navigate to the 'System' section.**

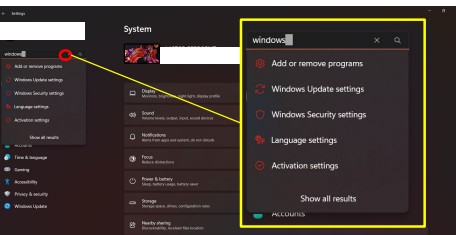

**Error:** The dropdown menu hides the "System" button. Thus the GUI Parser and MLLM cannot find the "System" in this screenshot.

**Query 2: Set the brightness under Dynamic Lighting with 70%**

**Subtask: Drag to set brightness to 70%**

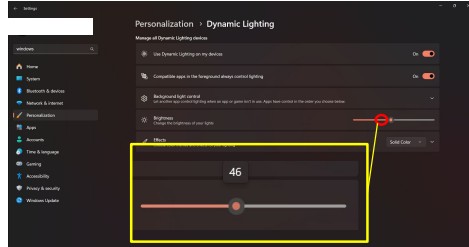

**Error:** The Actor cannot provide exact bbox to achieve the goal of dragging to 70%.

Figure 25: We display some common errors.

**Query 1: Format the slide background with gradient fill**

**Subtask: Select 'Center'**

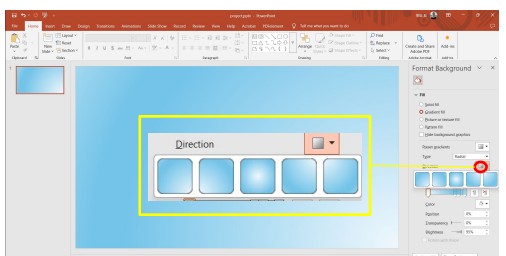

**Error:** The Actor Model cannot identify the "center" button in this dropdown menu

**Query 2: Replace all the 'Reading' in the text with 'RA'**

**Subtask: Enter 'RA' in the field behind the 'Replace with' text.**

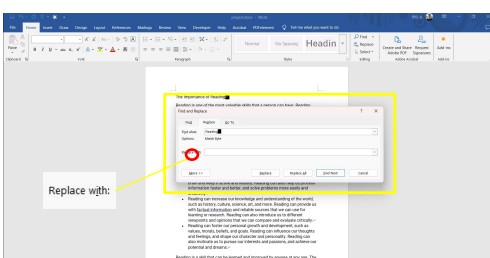

**Error:** The Actor cannot click on the input field as the parsed bbox is the position of text "Replace with"

Figure 26: We display some common errors

