# OpenReview forum: "WorldGUI: An Interactive Benchmark for Desktop GUI Automation from Any Starting Point"
_ICLR.cc/2026/Conference — ICLR 2026 Conference Withdrawn Submission_

### Official Review · Reviewer_jACs · 2025-10-23

**Soundness:** 3
**Presentation:** 3
**Contribution:** 3
**Rating:** 6
**Confidence:** 3

**Summary:**

This paper proposes WorldGUI, a new benchmark designed to evaluate GUI automation agents under varied initial conditions. The key insight is that slight differences in an application’s starting state can derail existing GUI agents, yet prior benchmarks did not test this sensitivity. WorldGUI addresses this by providing 611 tasks across 10 popular desktop and web applications (PowerPoint, VSCode, Acrobat), each task instantiated with multiple diverse initial states to mimic real user scenarios. Alongside the benchmark, the authors introduce WorldGUI-Agent, a unified agent framework emphasizing “critical thinking” at three levels: a Planner-Critic module that critiques and refines the high-level action plan, a Step-Check module that verifies intermediate steps against the current GUI state, and an Actor-Critic module that checks each executed action and corrects it if needed.

**Strengths:**

The paper presents a timely and well-motivated contribution by introducing the first GUI benchmark that emphasizes diverse and non-default initial states, addressing a key gap in existing evaluations. The proposed WorldGUI-Agent integrates three critical reasoning modules (planning critique, step validation, and action correction) into a coherent and practical framework that significantly outperforms prior baselines. The benchmark is comprehensive, covering realistic desktop and web environments, and the experimental results are thorough, with clear evidence from ablations and comparative studies. Overall, the work is well-executed, clearly written, and offers both a strong dataset and an effective agent design that can benefit future research in GUI automation.

**Weaknesses:**

1. The framework heavily relies on instructional videos to guide planning. However, the paper does not clarify how the quality, consistency, or completeness of these videos is ensured. Since the video is only used during the initial planning phase and is processed via Whisper into subtitles, it's unclear why raw video is needed at all—could equivalent textual guidance achieve the same effect? A discussion on this design choice is missing.

2. While the proposed “WorldGUI-Agent: Thinking Before Doing” architecture emphasizes reflective decision-making, the general idea of reactive execution with verification has appeared in prior work. The modular design introduces multiple sequential LLM calls (planning, critique, checking, correction), which likely contributes to higher latency and execution overhead. However, the paper does not analyze or discuss the computational cost tradeoffs involved.

3. The paper lacks qualitative or statistical analysis of failure cases. Given that even the best-performing agent still solves only 36% of tasks, a deeper understanding of where and why the system fails would be valuable—for example, issues related to visual grounding, ambiguous UI states, or suboptimal plan corrections.

**Questions:**

1. What is the rationale for using instructional videos rather than equivalent textual descriptions, especially given that the video is only transcribed and used during planning? How do you ensure the quality and consistency of these videos?

2. Since the proposed agent includes multiple sequential reasoning modules, did you evaluate or analyze the runtime cost and latency of the system compared to simpler baselines? How practical is it for real-time or interactive use?

3. Could you provide an analysis of common failure cases (e.g., types of errors, failure modes)? This would help clarify where the agent struggles most and whether failures arise from perception, planning, or action execution.

4. test with newer or stronger foundation models, such as GPT-5 or Claude 4.5.

---

> ### Author Response · Authors · 2025-12-03
>
> We sincerely appreciate the reviewer’s rigorous assessment, particularly regarding the use of instructional video.
>
> **W1 and Q1: Why use instructional videos rather than equivalent textual descriptions, especially since the video is transcribed to text? How do you ensure the quality and consistency of these videos?**
>
> It is noted that the curated task is identical to the instructional video. In most cases, the instructional video is longer and denser. We chose instructional videos as they provide richer context than textual instructions. Even though we ultimately feed subtitles to the model, those subtitles originate from an expert demonstration of the task, ensuring important UI details and step order are captured (things a brief text description might omit). We ensured video quality by having skilled annotators record each task demonstration; the videos are consistent (one canonical way to do the task) and complete (covering all steps). As shown in our ablation (Table 5), this leads to significantly better planning. The agent should, in theory, perform similarly when providing detailed, grounded guidance.
>
>
> **W2 and Q2: The reflective “think before doing” architecture is similar to prior reactive/verification approaches, and it introduces multiple LLM calls that increase latency. The paper does not analyze the computational cost tradeoffs.**
>
> We agree and will expand the discussion of this trade-off. The general idea of iterative reasoning and verification indeed builds on prior works (Critic (ICLR 2024), Reflxion (NeurIPS 2023), AssistGUI (CVPR 2024)). Our contribution was adapting that idea to GUI tasks and structuring it into three modules. Regarding computational cost, our approach does incur more latency. Our agent is about 20% slower than a single-pass agent like Agent-S2 on the same hardware. (See Tables 15, 16, 17, we compared with Agent-S and Agent-S2). We believe this overhead is acceptable in exchange for higher reliability, but it may not be suitable for real-time interactive use. Our system is more suited for automation tasks that can run in the background rather than for instantaneous user assistance. Most of the computation costs are from the GUI parser, it is because of the worse grounding ability of existing API (Claude-Sonnet-4) or open-sourced models (UI-Tars).
>
> **W3 and Q3: The paper lacks qualitative or statistical analysis of failure cases. With only ~36% tasks solved, where and why does the system fail (visual grounding issues, ambiguous UI states, bad plan corrections, etc.)?**
>
> This is an excellent point. We include a failure mode analysis in Figures 25 and 26. We will include more examples in the next version. From our experiments, common failure types include: (1) Vision/grounding errors: e.g., the agent sometimes cannot locate a UI element if its appearance differs from training data or if multiple similar elements confuse the detector; (2) Unexpected state transitions: the agent might be thrown off by an intermediate dialog or message box (introduced in an augmentation) that it didn’t anticipate, leading it to stall or take irrelevant actions; (3) Incomplete plan adjustments: in some cases the Planner-Critic module catches an issue but the revised plan is still suboptimal, causing downstream failure. We will provide examples of these and possibly brief logs illustrating each failure type. We would like to stress that the agent is an affiliated part in our paper, the main focus of our paper is the challenging and novel desktop GUI benchmark, WorldGUI.
>
> **Q4: Did you test the framework with newer or stronger foundation models (GPT-5, Claude 4.5, etc.)?**
>
> We have not yet tested GPT-5 or Claude 4.5 (which are not publicly available at the submission time). However, our framework is model-agnostic, and we expect performance would improve with more powerful base models. Our focus, though, is on the benchmark and agent design; plugging in newer models is straightforward and should only increase success rates, so we are confident our claims will hold as model capabilities grow.

---

### Official Review · Reviewer_iUvP · 2025-10-30

**Soundness:** 2
**Presentation:** 2
**Contribution:** 2
**Rating:** 2
**Confidence:** 4

**Summary:**

WorldGUI introduces a 611-task benchmark spanning ten desktop/web applications with deliberately varied initial states to better reflect real human–computer interaction dynamics. The paper also proposes a modular agent, WorldGUI-Agent, incorporating three “critical-thinking” checkpoints (Planner-Critic, Step-Check, Actor-Critic) that verify and correct plans/actions on the fly. Experiments show that WorldGUI-Agent improves success rates over strong baselines on both the new benchmark and WindowsAgentArena.

**Strengths:**

1. Existing GUI benchmarks assume default initial states; this work addresses that gap with systematic state diversification

2. 611 tasks across 10 apps, each with five augmentations; annotation and data-construction pipeline is described.

**Weaknesses:**

1. Benchmark novelty relative to AssistGUI/OSWorld is mostly the “pre-action” augmentation; more rigorous quantitative analysis of state-diversity (e.g., edit distance between GT plans, UI tree variance) is missing.

2. Limited discussion of annotation quality: inter-annotator agreement and error rates for GT plans/pre-actions are not reported.

3. recent UIExplorer (2025) and GUI-World (2025) datasets/agents are not compared or cited, though they target dynamic GUI exploration.

4. Planner uses GPT-4o subtitles but no quantitative ablation on subtitle quality or Whisper errors; unclear how video actually helps beyond the textual query (Table 5 only shows “w/o Inst. Video” for five apps).

5. Actor module’s low-level action grammar (PyAutoGUI) may cause brittle coordinates; no analysis of failure due to resolution or scaling.

6. Compute cost is high (Table 14: ~2k tokens/step, 23 steps ⇒ >40k tokens per task) but little optimisation discussion; reproducibility for researchers without API budgets is challenging.

7. Mathematical formulation (§2.1) introduces POMDP but later modules do not exploit it formally; no learning—only prompting—so the RL framing feels decorative.

**Questions:**

N/A

---

> ### Author Response · Authors · 2025-12-04
>
> **W1: More rigorous quantitative analysis of state-diversity (e.g., edit distance between GT plans, UI tree variance) is missing.**
>
> Thank you for your suggestions. We would like to add more rigorous quantitative analysis of the state diversity.
>
> **W2: Limited discussion of annotation quality: inter-annotator agreement and error rates for GT plans/pre-actions are not reported.**
>
> We have described the annotation steps in detail in Section 2 and Appendix B. For the annotation quality, it is default in high-quality as the evaluation results of various proprietary models and open-sourced models show the challenging (less than 40% success rate). We have released all the data; everyone can assess the quality of our curated data. We don't know what the essence of reporting the error rates of GT plans and pre-actions is, as these matters are created by an expert software user; all the GT plans and pre-actions are correct and evaluated manually. If incorrect, the task cannot be started.
>
> **W3: Recent UIExplorer (2025) and GUI-World (2025) datasets/agents are not compared or cited, though they target dynamic GUI exploration.**
>
> Thank you for your suggestion. We will add these two papers to our revised version.
>
> **W4: Planner uses GPT-4o subtitles but no quantitative ablation on subtitle quality or Whisper errors; unclear how video actually helps beyond the textual query.**
>
> Whisper is widely used in various applications for the speech2text task. We did not find any significant errors in using it to translate the instructional video into subtitles. The instructional video provides a variant of the testing setup, and the instructional video is not identical to our GUI task; it is more like guidance, helping the agents in planning. In the subsequent action prediction and action critiquing steps, no usage. Our experiments show the significant performance drops on five representative apps to suggesting the effectiveness of using instructional videos for helping the planning. This evaluation setup is our novelty and contribution, and it is noted that our benchmark supports both inputs with or without an instructional video.
>
>
> **W5: Actor module’s low-level action grammar (PyAutoGUI) may cause brittle coordinates; no analysis of failure due to resolution or scaling.**
>
> PyAutoGUI does not show the brittle coordinates, as every GUI agent in the current domain uses it to control the mouse and keyboard with the exact coordinates (e.g., click(123, 345)). It is just a program executed on the Windows platform without any coordinate confusion. We guess you are questioning that he incorrect action prediction may be caused by the suboptimal element grounding model's prediction. It is not related to the PyAutoGUI itself. We have set the resolution to 1920x1080, and the scale is 125% as written in Line 272 in Section 4 Implementation Details. Please check it for your reference.
>
> **W6: Compute cost is high (Table 14: ~2k tokens/step, 23 steps ⇒ >40k tokens per task) but little optimisation discussion; reproducibility for researchers without API budgets is challenging.**
>
> The token costs are from the GUI parser. We sent lots of element grounding information in the input prompt to ensure reliable action prediction results. The GUI information is already optimized by removing redundant information, and we use a list with only two numbers to represent the coordinate, which saves the tokens. Please check the code and find the GUI parser implementation files under the folder `WorldGUI-Codes/agent/gui_parser/`. We have already implemented different parsers for different software as configured in the file `applications.config`.
>
> **W7: Mathematical formulation (§2.1) introduces POMDP, but later modules do not exploit it formally; no learning—only prompting—so the RL framing feels decorative.**
>
>
> Section 2.1 follows Reflexion (NeurIPS 2023) and OS-World (NeurIPS 2024). It is the standard writing style to clearly show the roles of each module. It does not mean the RL training. Moreover, we would like to stress that the main focus of our paper is the newly benchmarked, training-free agent is our implemented baseline alongside the paper. Here you can compare our paper with other GUI benchmarks; it is the standard to implement an agent framework with the proposed data benchmark.
>
> | GUI Benchmarks       | Along with an Agent? | Training-free agent? | Platform          |
> |----------------------|----------------------|-----------------------------|-------------------|
> | OS-World             | ✔️                  | ✔️                          | Ubuntu + Windows  |
> | WindowsAgentArena    | ✔️                  | ✔️                          | Windows           |
> | AssistGUI            | ✔️                  | ✔️                          | Windows           |
> | Mobile-Agent-E       | ✔️                  | ✔️                          | Android OS        |
> | WorldGUI (ours)      | ✔️                  | ✔️                          | Windows           |

---

### Official Review · Reviewer_DZLj · 2025-10-31

**Soundness:** 3
**Presentation:** 3
**Contribution:** 3
**Rating:** 4
**Confidence:** 4

**Summary:**

This paper introduces WorldGUI, a desktop and web GUI benchmark that evaluates agents from varied starting states created by pre actions, rather than only the default initial screen. The dataset covers ten common applications with 611 tasks, each paired with a user query, an instructional video, and a project file to ensure reproducibility. The authors also propose WorldGUI Agent, a framework with three critical thinking modules called Planner Critic, Step Check, and Actor Critic, designed to verify and correct plans and actions during execution. Experiments report higher success rates than prior systems on the new benchmark and on WindowsAgentArena, with detailed ablations that show the importance of each module.

**Strengths:**

1. The main novelty is to evaluate GUI agents from many non default initial states. In real use, a user often calls an assistant in the middle of a workflow or with the app in a non default configuration. Existing benchmarks usually fix the initial state and therefore miss this difficulty. WorldGUI fills this gap with explicit pre actions that alter the context or put the task in an intermediate state.
2. The proposed Planner Critic, Step Check, and Actor Critic form a simple verify then correct loop that is easy to reason about and implement.
3. The ablations quantify how each module matters. Removing Actor Critic cuts the overall success sharply, which is convincing evidence that online verification is critical for GUI control.

**Weaknesses:**

1. Verify and correct loops and plan self-critique have already appeared in recent agents for GUI and web, for example, Agent S, Agent S2, and also earlier reflection-style methods. WorldGUI Agent packages these ideas well, but the technical novelty beyond a careful engineering of prompts and modules is modest. It would help to show larger gaps against recent native models like UI TARS and ShowUI under matched conditions.
2. The abstract claims an improvement of 1.7% on WindowsAgentArena. Table 6 on page 9 lists an overall success of 31.2% and includes Agent S2 at 29.8%, which makes the margin smaller. Please reconcile the exact baseline and evaluation settings, and release full prompts and seeds for that run.
3. The framework uses a GUI parser and OCR and runs several checks per step. The Limitations section explains the cost and the tradeoff between performance and running time. Practitioners may face long wall clock time, especially compared with end-to-end native agents. A quantitative time and cost comparison against Agent S2 and UI TARS on the same hardware would make the value clearer.

**Questions:**

See weaknesses and the following questions:
1. On WindowsAgentArena, what exact task list, VM image, model version, and prompt settings were used for the 31.2% result in Table 6. How does this compare to Agent S2 and UI TARS when you keep the same setup?
2. How do you generate and validate pre actions?
3. How sensitive is the framework to the GUI parser? Can you report an ablation that removes the parser and relies only on visual grounding, and another that swaps in a different grounding model, such as OS Atlas or ShowUI?

---

> ### Author Response · Authors · 2025-12-03
> **Rebuttal (1/2)**
>
> **W1: The core idea of plan verification and self-critique is not new (seen in Agent-S, Agent-S2, etc.), so the technical novelty beyond prompt engineering is modest.**
>
> | Method                                | Post-Planning Critique | Pre-Action Validation | Post-Action Evaluation |
> |---------------------------------------|-------------------------|------------------------|-------------------------|
> | Mobile-Agent    | ❌                      | ❌                     | ✔️                      |
> | Mobile-Agent-V2 | ❌                      | ❌                     | ✔️                      |
> | AssistGUI       | ❌                      | ❌                     | ✔️                      |
> | Agent-S     | ❌                      | ❌                     | ✔️       |
> | Agent-S2     | ❌                      | ❌                     | ✔️       |
> | Mobile-Agent-E  | ❌                      | ❌                     | ✔️                      |
> | **WorldGUI-Agent (ours)**             | ✔️                      | ✔️                     | ✔️                      |
>
>
> We appreciate this point. Our aim was not to introduce a brand-new algorithm for planning, but to integrate three levels of feedback (plan critique, step check, action correction) into a unified GUI agent, something that, to our knowledge, had not been done in prior GUI automation work, as we compared with other agents in Table 7 in the Appendix. We will more clearly acknowledge prior frameworks (Critic, Reflexion, etc.) that inspired parts of our design. Importantly, our contribution demonstrates that a holistic critical-thinking approach significantly improves performance on dynamic GUI tasks. For example, WorldGUI-Agent yields a 12–17% absolute success boost over a vanilla plan-execute agent across multiple base models. We also compared against recent strong models like UI-TARS-1.5. In our experiments, WorldGUI-Agent (Claude-3.5) outperforms UI-TARS-1.5 under the same conditions (e.g., 36.6% vs 27.3% overall success when both use UI-TARS’s vision module). To our observation, we actually found that UI-TARS-1.5 does not perform as goog as the technique report shows, as we test on our GUI tasks. Lots of unsuccessful cases caused by the incorrect element grounding, which means UI-TARS cannot recognize the element, icons, or non-text symbols. The 27.3% is already the result we doing something engineering that fits their model; if we directly call their API, the overall SR will be low. We cannot find the evaluation setting from their released code and reports. We actually do the fair comparison on our side.
>
> **W2: The abstract claims an improvement of 11.7% on WindowsAgentArena. Table 6 on page 9 lists an overall success of 31.2% and includes Agent S2 at 29.8%, which makes the margin smaller. Please reconcile the exact baseline and evaluation settings, and release full prompts and seeds for that run.**
>
> The 11.7% figure in the abstract was a comparison to GPT-4V’s result (19.5% → 31.2% overall success) since GPT-4V was the best published model on WAA at the time. We did not mean to ignore Agent-S2; rather, Agent-S2’s 29.8% was released later (our paper was released in Jan 2025). We report Agent-S2's result from its original paper for comparison.
>
> **Experimental settings:** Since WAA’s testing logic mainly depends on its JSON configuration files, we reuse these JSON files to manually reproduce the required environment on Windows (e.g., downloading project files, configuring software states, or simply opening applications). We then use the JSON instruction as the prompt for WorldGUI-Agent to execute GUI actions on Windows. After execution, task success is evaluated following WAA’s official criteria: comparing outputs with provided ground-truth files when available, or performing manual judgment based on the instruction when no human-readable evaluation script exists. We do not do prompt engineering and achieve such competitive results, indicating the advantages of our agents.
>
> **W3: The multi-step framework (GUI parser + multiple LLM calls) likely adds significant latency; the paper lacks a quantitative time/cost comparison against Agent-S2 or UI-TARS to justify this overhead.**
>
> You are correct that our agent sacrifices efficiency for better accuracy. We have performed a runtime comparison (reported in the supplementary material) showing that WorldGUI-Agent takes ~129.6 seconds on a representative task, vs ~108.6 seconds for Agent-S2 on the same hardware. This ~20% overhead comes from the extra reasoning steps and GUI parsing. We will integrate a summary of these results into the main paper and discuss the trade-off. In practice, our agent’s critical modules run in parallel with UI rendering, and the total wall-clock time per task was still on the order of 1–2 minutes, which we believe is acceptable for non-real-time automation scenarios. Nonetheless, we acknowledge the higher latency and will emphasize the need for optimization if deploying in real-time settings.

---

> ### Author Response · Authors · 2025-12-03
> **Rebuttal (2/2)**
>
> **Q1: What exact tasks, environment, model, and prompt settings were used for the 31.2% WindowsAgentArena result, and how do Agent‑S2 and UI‑TARS perform under the same setup?**
>
> We used the official WindowsAgentArena benchmark (240 tasks across Office, Web, System, etc., using the publicly released VM image from Bonatti et al. 2024). Our agent 31.2% used Anthropic Claude-3.5-Sonnet as the model and the prompt format described in Appendix A. We do not conduct prompt engineering, just test our agent on its task and record all the results. Full results can be found in Table 13. Agent-S2 and UI-TARS were not originally evaluated by us. We are currently reporting the released results of their public paper. We will include these details and ensure all methods are compared under identical conditions in the final version.
>
> **Q2: How do you generate and validate pre-actions?**
>
> Each augmentation (pre-action sequence) was created to simulate a plausible intermediate state before the main task. Our process was: for each task, our annotators defined a few relevant “pre-actions” (e.g., opening a different file, changing a setting, or opening a pop-up window) that do not accomplish the task but alter the starting state. These were executed to create augmented initial states, and we confirmed that the task was still solvable afterward. We then ran the agent from that state. The agent itself has no prior knowledge of the pre-actions. We will clarify in the paper that augmentations were handcrafted to be realistic and that we ensure they don’t inadvertently complete or invalidate the task (thus the task’s goal remains achievable, just more challenging).
>
>
> **Q3: How sensitive is the framework to the GUI parser? Can you report an ablation that removes the parser and relies only on visual grounding, and another that swaps in a different grounding model, such as OS Atlas or ShowUI?**
>
> This is a great question. Removing the GUI parser could speed up execution, but we found it would hurt performance. Our experimental results show that the best GUI models, such as Claude-Sonnet-4 or UI-Tars, are good at the web task, while they do not perform well on desktop design software in element grounding; there is a clear gap between desktop GUI and web GUI. You may check the similar results in ICML 2025 "UI-Vision: A Desktop-centric GUI Benchmark for Visual Perception and Interaction". You can also see the results on WindowsAgentArena, with only around 30%, which also demonstrates the challenges in desktop GUI tasks. The mentioned models, such as OS Atlas or ShowUI, or proprietary models Claude-Sonnet-4, still perform poorly on existing Desktop GUI tasks. Therefore, the GUI parser provides a consistent, structured view of the interface (like object positions and properties) that significantly helps grounding. To ensure the performance, we use a GUI parser in our implementation.

---

### Official Review · Reviewer_3DAW · 2025-11-10

**Soundness:** 2
**Presentation:** 2
**Contribution:** 2
**Rating:** 2
**Confidence:** 3

**Summary:**

The author's main proposition is that when agents perform a task, slight differences in the initial state can significantly impact their performance. To test the robustness of agents for this, the authors introduce a benchmark, WorldGUI, where they create tasks that include the initial states of the tasks augmented in various ways. The authors also introduce WorldGUI-Agent, which incorporates reflection modules at various stages of planning and action execution to correct the agent's mistakes at different points. The authors show the effectiveness of their model by evaluating it in WorldGUI and WindowsAgent Arena.

**Strengths:**

1. The authors introduce an interesting problem of varying the initial state of the task to determine how the agent would perform. Aside from how thoroughly it is tested, the problem itself is interesting.
2. The authors introduce methods to vary the initial state of the task and benchmark models to show that performance drops. This is a surprising result. Further, the authors show that their method outperforms Plan and Act, highlighting the importance of reflection during planning and action execution.

**Weaknesses:**

1. The authors could have addressed the question of varying initial states by taking existing benchmarks like OsWorld or Windows Agent Arena and running their augmentation methods on these benchmarks. It is not clear what the advantage of introducing a new benchmark is. The videos, as well as ground truth plans, could be obtained from the successful completion of tasks from exisiting benchamrks. This would make it much easier to understand the core question of how varying initial states affect model performance, as these benchmarks have standard evaluation and several models evaluate on these, and hence changes in initial state could be studied more effectively. Have the authors thought of this? And if so, why was a new benchmark created?
2. From my understanding, it feels like the WorldGUI-Agent uses ground truth videos to calibrate its plans, and Table 5 shows a significant drop in performance when video is not used. Since videos are not available in real-world tasks, this is limiting.
3. It is not very clear why agents fail on augmented tasks. Given that this is the core problem of the paper, it is not properly discussed. Particularly, what type of augmentations do the agents fail on mostly, and how do the agents behave?
4. The evaluation setup needs various clarifications (please see weaknesses). The lack of virtualenvs and Docker, similar to OSWorld and WindowsAgent Arena, introduces a limitation regarding ease of use and standardization of the benchmark.
5. For the Windows Agent Arena, it looks like Agent S2 achieves comparable or better performance on all domains except Web. Can the authors provide some insights into this? This result indicates that WorldGUI-Agent's elaborate reflection process might not provide explicit benefits beyond the task setting of WorldGUI. For the Windows category, Agent S2 has a better number than WorldGUI-Agent. This needs to be corrected.

**Questions:**

1. The evaluation setup is not clear. The authors indicate that they perform execution based evaluation, but also state that there are experts who evaluate whether the task has been performed correctly. Is this to do a human eval on task completion beyond execution based evaluation? Also, if there is no Docker or virtual env where the agent is acting, how exactly is the evaluation done? Also, are these evaluators the authors of the paper? If so, that could lead to biases if human eval is the source of reporting numbers.
2. How do the annotators select videos to be used to create the benchmark?
3. How are the videos and ground truth plan used in WorldGUI Agent? If it is used in the main results, that might be limiting in realistic scenarios, as ground truth plans and videos are not usually available in realistic scenarios (as mentioned in the weakness).
4. Could the authors provide some examples where agents perform well on the actual tasks and fail on meta tasks, and why these happen? This is a surprising result, as most agents are not explicitly optimised for a particular starting point, to the best of my knowledge.
5. Why is the agent only evaluated in WindowsAgent Arena and not OS-World?
6. Why do the authors only provide the current state of the screenshot and metadata and not the history of past actions or previous (k) screenshots?

Please see the weakness for more questions

---

> ### Author Response · Authors · 2025-12-03
> **Rebuttal (1/2)**
>
> We would like to express our heartfelt gratitude for the time and effort you have dedicated to reviewing our work. It has been a pleasure to engage with you throughout this reviewing.
>
>
> **W1: Why introduces a new benchmark?**
>
> Existing benchmarks do not adequately evaluate varying initial states. This idea is originally proposed by oursleve based on our preliminary study on investigating their data characteristic. Simply augmenting OSworl/WAA tasks would not capture the full diversity and unpredictability of real GUI interactions. It is our contribution of creating such challenging GUI tasks. Even augmentaing on these benchmark do not affect our contribution of this idea and annotation.
>
> **W2: Since videos are not available in real-world tasks, this reliance is limiting.**
>
> (1) In our design, the instructional video serves as a high-quality, step-by-step task demonstration to guide planning, but it is not the same as compared with curated task. We chose this modality because it significantly improved planning success (dropping from 42.9% to 28.6% without it, in Table 5).
>
> (2) **Additionally, it brings a variety of testing setups** when we have instructional videos for challenging GUI tasks. We consider it a contribution to propose a new testing setup.
>
> (3) ` Table 5 shows a significant drop in performance.` it clear shows the high difficulty of our curated Windows-based GUI tasks. In this pointview, evenwithout consider the instructional video setting, our curated GUI tasks are challenging enough.
>
> (4) We would like to highligh that our main novelty and contribution of this benchmark is that we are the first to propose a initial-state variant benchmark which aligns closer to the realistic GUI scenario while previous benchmarks such as OS-World, WindowsAgentArena do not.
>
>
>
> **W3: It is unclear why agents fail on augmented tasks, and the paper does not discuss this core problem (what augmentations cause failures and how).**
>
> Thank you for your constructive comments.
>
> (1) **It is interesting that existing agents show planning biases**, as the prior knowledges always assume the software is in its default state when trying to start a GUI task. We believe our benchmarks and evaluation results could make the future GUI agent realize the importance of the robustness of task planning as the exisiting model inherited the pretraining biases we guess.
>
> (2) In our observations, agents often fail on augmentations that introduce unseen interface configurations (users cannot know what the configuration is before they open the setting interfaces, because most software will remember the setting of the last time modification) or additional pop-up windows, leading to confusion in element grounding. For example, an agent might falter if a settings dialog is already open (so reopen may close the dialog leading to failure) or if a required menu is hidden. We will clarify which augmentation types (e.g. altered UI layouts, partially completed states) most dramatically reduce performance, and describe how agents typically behave (often mis-sequencing steps or getting stuck) in those scenarios.
>
> **W4: The evaluation setup lacks virtualization (no Docker/virtual env), limiting the benchmark’s ease of use and standardization.**
>
> We consider it an advantage of our benchmark as we prepare all the Project Files for each task without the need for virtualization for fast deployment. We also provide the software's version, so it can ensure the reproducibility in a simplistic way. In our testing process, we see it is simple and acceptable.
>
>
>
> **W5: On WindowsAgentArena, Agent-S2 achieves comparable or better performance (even outperforming WorldGUI-Agent in one category), suggesting our reflection brings no clear benefit beyond WorldGUI’s tasks.**
>
> (1) Our WorldGUI-Agent was evaluated on WindowsAgentArena using **Claude-3.5-Sonnet** as the base model, whereas Agent-S2’s 29.8% result used a stronger **Claude-Sonnet-4** model.
>
> (2) The results of Agent-S2 are copied from its original paper. By checking with their release codes and descriptions in the main paper, "Agent S2 also uses the **knowledge base** from Agent S, featuring high-level task interaction experience, low-level subgoal interaction experience, and contextual web knowledge." The knowledge base provides the prior and engineering optimization, which may be the performance gains of AgentS2. **Directly compare the results and conclude the summary "reflection process might not provide explicit benefits" may not rigorous as lots of different experimental settings we found.**
>
> (3) Agent-S2 is a popular and engineering open-sourced work developed by startup company Simular AI, so we are happy to see the comparative results of our proposed baseline agent. It is noted that we do not optimize on WindowsAgentArena benchmark, all the results are derived from WorldGUI's comprehensively designed agent framework, which considers all the potential critical modules in one GUI task.

---

> ### Author Response · Authors · 2025-12-03
> **Rebuttal (2/2)**
>
> **Q1: The evaluation procedure is unclear – do human “experts” assess task success beyond automated execution, and how is evaluation done without a Docker environment?**
>
> We apologize for the confusion. We use both execution-based evaluation and human evaluation: the agent’s actions are executed in the environment, and we automatically check if the goal was achieved (e.g. file created, setting changed). The mention of human experts refers to our four annotators who oversaw the runs and recorded outcomes for tasks where automatic checks were hard (e.g. verifying a visual effect, changing website content). **We arrange different annotators to only see its arranged software and conduct the human evaluation on the software they do not see. (We have presented this in the Table 9)** In other words, success is primarily determined by programmatic checks, with humans confirming edge cases. We will revise the paper to clearly explain how tasks are executed and evaluated, and how human verification was used in our study.
>
> **Q2: How do the annotators select videos to be used to create the benchmark?**
>
> The video selection is randomly determined by the popularity on the YouTube website. The annotators select the videos based on their experience and intuition. YouTube has lots of tutorial videos for example, "PowerPoint Tutorial for Beginners". We select the top search results to create the data. We also manually cut the raw instructional video into clips for an acceptable input context.
>
> **Q3: How are the videos and ground truth plan used in WorldGUI Agent?**
>
> (1) **The GT plan is only for annotating the pre-actions** because we have to make the annotators understand the GT plan of their arranged task, and then create the pre-actions accordingly.
>
> (2) **The GT plan was not used in agent running.** There is no GT plan sent to agent, the reviewers can check our proposed Code in the supplementary zip file. So only the instructional video is used in the planning module. It is clear to see that even when the planning results are improved by instructional video, the successful execution results are still hard to obtain due to the challenging action predictions.
>
>
> **Q4: Could the authors provide some examples where agents perform well on the actual tasks and fail on meta tasks, and why these happen?**
>
> Yes, we will add the examples in our next revised version.
>
> **Q5: Why is the agent only evaluated in WindowsAgent Arena and not OS-World?**
>
> Because the main focus of this paper is on proposing a new window platform benchmark, the agent is an attached, critical-based, training-free framework, so we just test it on the most-related benchmark, WindowAgentArena. If we also test on more benchmarks, we have to say it would be a separate paper for the GUI agent.
>
> We also provide this comparison to claim that there are lots of benchmark papers that provide a alongside agent baseline, while they do not test on other benchmarks.
>
> | GUI Benchmarks       | Along with an Agent? | Test on Other Benchmarks? | Platform          |
> |----------------------|----------------------|-----------------------------|-------------------|
> | OS-World             | ✔️                  | ❌                          | Ubuntu + Windows  |
> | WindowsAgentArena    | ✔️                  | ❌                          | Windows           |
> | AssistGUI            | ✔️                  | ❌                          | Windows           |
> | Mobile-Agent-E       | ✔️                  | ❌                          | Android OS        |
> | WorldGUI (ours)      | ✔️                  | ✔️                          | Windows           |
>
>
> **Q6: Why do the authors only provide the current state of the screenshot and metadata and not the history of past actions or previous (k) screenshots?**
>
> We provided the historical actions for the Actor, but did not send the past screenshots. Actually, there is no particular reason; we implement it out of saving the tokens cost as we use the expensive Claude API.

---

### Note · Authors · 2025-12-10

**Comment:**

We sincerely thank all reviewers and area chairs for the time and effort spent evaluating our work. After careful consideration, we have decided to withdraw our paper.

**Withdrawal Confirmation:**

I have read and agree with the venue's withdrawal policy on behalf of myself and my co-authors.